# Toward an open-access of high-frequency lake modelling and statistics data for scientists and practitioners. The case of Swiss Lakes using Simstrat v2.1

Adrien Gaudard[1†], Love Råman Vinnå[1], Fabian Bärenbold[1], Martin Schmid[1], Damien Bouffard[1]

[1]Surface Waters Research and Management, Eawag, Swiss Federal Institute of Aquatic Sciences and Technology, Kastanienbaum, Switzerland

[†] deceased, 2019

*Correspondence to*: Damien Bouffard (damien.bouffard@eawag.ch)

**Abstract**

One-dimensional hydrodynamic models are nowadays widely recognized as key tools for lake studies. They offer the possibility to analyse processes at high frequency, here referring to hourly time scale, to investigate scenarios and test hypotheses. Yet, simulation outputs are mainly used by the modellers themselves and often not easily reachable for the outside community. We have developed an open-access web-based platform for visualization and promotion of easy access to lake model output data updated in near real time (simstrat.eawag.ch). This platform was developed for 54 lakes in Switzerland with potential for adaptation to other regions or at global scale using appropriate forcing input data. The benefit of this data platform is practically illustrated with two examples. First, we show that the output data allows for assessing the long term effects of past climate change on the thermal structure of a lake. The study confirms the need to not only evaluate changes in all atmospheric forcing but also changes in the watershed or through-flow heat energy and changes in light penetration to assess the lake thermal structure. Then, we show how the data platform can be used to study and compare the role of episodic strong wind events for different lakes on a regional scale and especially how their thermal structure is temporarily destabilized. With this open-access data platform we demonstrate a new path forward for scientists and practitioners promoting a cross-exchange of expertise through openly sharing of in-situ and model data.

## 1 Introduction

Aquatic research is particularly oriented towards providing relevant tools and expertise for practitioners. Understanding and monitoring inland waters is often based on *in situ* observations. Today, the physical and biogeochemical properties of many lakes are monitored using monthly to bi-monthly vertical discrete profiles. Yet, part of the dynamics is not captured at this temporal scale (Kiefer et al., 2015). An emerging alternative approach consists in deploying long-term moorings with sensors and loggers at different depths of the water column. However, this approach is seldom used for country-level monitoring, although it is promoted by research initiatives such as GLEON (Hamilton et al., 2015) or NETLAKE (Jennings et al., 2017).

It is common to parameterize aquatic physical processes with mechanistic models, and ultimately use them to understand aquatic systems through scenario investigation or projection of trends in for example a climate setting. In the last decades, many lake models have been developed. They often successfully reproduce the thermal structure of natural lakes (Bruce et al., 2018). Today's most widely referenced one-dimensional (1D) models include (alphabetic order) DYRESM (Antenucci and Imerito, 2000), FLake (Mironov, 2005), GLM (Hipsey et al., 2014), GOTM (Burchard et al., 1999), LAKE (Stepanenko et al., 2016), Minlake (Riley and Stefan, 1988), MyLake (Saloranta and Andersen, 2007), and Simstrat (Goudsmit et al., 2002). The results from these models are mainly used by the modellers themselves, and often not easily accessible for the outside community.

The performance of lake models is determined by the physical representativeness of the algorithms and by the quality of the input data. The latter include (i) lake morphology, (ii) atmospheric forcing, (iii) hydrological cycle (e.g. inflow, outflow and/or water level fluctuations), and (iv) light absorption. *In situ* observations, such as temperature profiles, are required for calibration of model parameters. To support this approach, it is important to promote and facilitate the sharing of existing datasets of observations among scientists and practitioners. Conversely, scientists and practitioners should benefit from the model output, which is often ready-to-use, high-frequency and up-to-date. Yet, model output data should not only be seen as a tool for temporal interpolation of measurements. Models also provide data of hard to measure quantities which are helpful for specific analyses (e.g., the heat content change to assess impact of climate change, or the vertical diffusivity to estimate vertical turbulent transport). Models finally support the interpretation of biogeochemical processes which often depend on the thermal stratification, mixing and temperature. In a global context of open science, collaboration between the different actors and reuse of field and model output data should be fostered. Such win-win collaboration serves the interests of lake modellers, researchers, field scientists, lake managers, lake users, and the public in general.

In this work, we present a new automated web-based platform to visualize and distribute the near real time (weakly) output of the one-dimensional hydrodynamic lake model Simstrat through an user-friendly web interface. The current version includes 54 Swiss lakes covering a wide range of characteristics from very small volume such as Inkwilersee ($9 \times 10^{-3}$ km$^3$) to very large systems such as Lake Geneva (89 km$^3$), over an altitudinal gradient (Lago Maggiore at from 193 m. a.s.l. to Daubensee at 2207 m. a.s.l.) and over all trophic states (14 euthrophic lakes, 10 mesotrophic lakes and 21 oligotrophic lakes, Appendix

A). We focus here on describing the fully automated workflow, which simulates the thermal structure of the lakes and weekly updates the online platform (https://simstrat.eawag.ch) with metadata, plots and downloadable results. This state-of-the-art framework is not restricted to the currently selected lakes and can be applied to other systems or at global scale.

## 2 Methods

### 2.1 Model and workflow

We use the 1D lake model Simstrat v2.1 to model 54 Swiss lakes or reservoirs (see Appendix A for details of modelled lakes) in an automated way. Simstrat was first introduced by Goudsmit et al. (2002) and has been successfully applied to a number of lakes (Gaudard et al., 2017; Perroud et al., 2009; Råman Vinnå et al., 2018; Schwefel et al., 2016; Thiery et al., 2014). Recently, large parts of the code were refactored using the object-oriented Fortran 2003 standard. This version of Simstrat provides a clear, modular code structure. The source code of Simstrat v2.1 is available via GitHub at: https://github.com/Eawag-AppliedSystemAnalysis/Simstrat/releases/tag/v2.1. A simpler build procedure was implemented using a docker container. This portable build environment contains all necessary software dependencies for the build process of Simstrat. It can therefore be used on both Windows and Linux systems. A step-by-step guide is provided on GitHub.

In addition to the improvements already described by Schmid and Köster (2016), Simstrat v2.1 includes (i) the possibility to use gravity-driven inflow and a wind drag coefficient varying with wind speed – both described by Gaudard et al. (2017), and (ii) an ice and snow module. The ice and snow module employed in the model is based on the work of Leppäranta (2014, 2010) and Saloranta and Andersen (2007), and is further described in Appendix B.

A Python script was developed to (i) retrieve the newest forcing data directly from data providers and integrate them into the existing datasets, (ii) process the input data and prepare the full model and calibration setups, (iii) run the calibration of the model for the chosen model parameters, (iv) provide output results, and (v) update the simstrat.eawag.ch online data platform to display these results. The script is controlled by an input file written in JSON format, which specifies the lakes to be modelled together with their physical properties (depth, volume, bathymetry, etc.) and identifies the meteorological and hydrological stations to be used for model forcing. The overall workflow is illustrated in Figure 1.

### 2.2 Input data

Table 1 summarizes the type and sources of the data fed to Simstrat. For meteorological forcing, homogenized hourly air temperature, wind speed and direction, solar radiation and relative humidity from the Federal Office of Meteorology and Climatology (MeteoSwiss, CH) weather stations are used. For each lake the closest weather stations are used. Air temperature is corrected for the small altitude difference (see Appendix A) between the lake and the meteorological station, assuming an adiabatic lapse rate of -0.0065 °C m$^{-1}$. This correction is a source of error in high altitude lakes like Daubensee for which dedicated meteorological station would be needed. The cloud cover needed for downwelling longwave radiations are estimated

by comparing observed and theoretical solar radiation (Appendix C). For hydrological forcing, homogenized hourly data from the stations operated by the Federal Office for the Environment (FOEN) are used. For each lake, the data from the available stations at the inflows are aggregated to feed the model with a single inflow. The aggregated discharge is the sum of the discharge of all inflows, and the aggregated temperature is the weighted average of the inflows for which temperature is measured. Inflow data are often missing for small or high altitude lakes (Appendix A). Missing inflows and more generally watershed data is a source of error in small alpine lakes, yet, such error can be compensated during the calibration process. The light absorption coefficient $\varepsilon_{abs}$ [m$^{-1}$] is either obtained from Secchi depth $z_{Secchi}$ [m] measurements (for Inkwilersee, Lake Biel, Lake Brienz, Lake Geneva, Lake Neuchatel, Lower Lake Zurich, Oeschinensee, Upper Lake Constance, and Sihlsee), or is set to a constant value based on the lake trophic status. In the first case, the following equation is applied: $\varepsilon_{abs} = 1.7/z_{Secchi}$ (Poole and Atkins, 1929, Schwefel et al. 2016). In the second case, $\varepsilon_{abs}$ is set to 0.15 m$^{-1}$ for oligotrophic lakes, 0.25 m$^{-1}$ for mesotrophic lakes, and 0.50 m$^{-1}$ for eutrophic lakes. The values correspond to observations of Secchi depths in Swiss lakes (Schwefel et al. 2016) and fall into the decreasing range of transparency from an oligotrophic to eutrophic system (Carlson 1977). For glacier-fed lakes (typical above 2000 m) rich in sedimentary material, $\varepsilon_{abs}$ is set to 1.00 m$^{-1}$.

The timeframe of the model is determined by the availability of the meteorological data (air temperature, solar radiation, humidity, wind, precipitation). Initial conditions for temperature and salinity are set using conductivity-temperature-depth (CTD) profiles or using the temperature information from the closest lake. We apply different data patching methods to remove data gaps from the forcing depending on the length of the data gap. For small data gaps with duration not exceeding one day, the dataset is linearly interpolated. In total < 1 % of the dataset is corrected using this approach. Longer data gaps of up to 20 days are replaced by the long-term average values for the corresponding day of the year. Only ~ 1.5 % of the dataset is corrected using this approach.

**2.3 Calibration**

Model parameters are set to standard default values, and four of them are calibrated (see Table 2). The parameters *p_radin* and and *f_wind* scale the incoming long-wave radiation and the wind speed, respectively, and can be used to compensate for systematic differences between the meteorological conditions on the lake and at the closest meteo station. The parameter *a_seiche* determines the fraction of wind energy that feeds the internal seiches. This parameter is lake-specific, as it depends on the lake's morphology and it's exposition to different wind directions. Finally, the parameter *p_albedo* scales the albedo of ice and snow applied to incoming shortwave radiation, which depends on the ice/snow cover properties and is unknown for the individual lakes. The calibration parameters were selected according to their importance for the model (e.g. based on previous sensitivity analysis), and their number was deliberately kept small in order to keep the calibration process simple and focused. Calibration is performed using PEST v15.0 (see http://pesthomepage.org), a model-independent parameter estimation software (Doherty, 2016). As a reference for calibration, temperature observations from CTD profiles are used. Calibration is performed on a yearly basis, unless significant changes are made to either the model, the forcing data, or the observational

data (e.g. release of a new version of Simstrat or delivery of a large amount of new observational data). For the eight lakes without observational data, parameters are set to their default value (see Table 2) with no calibration performed, and the lack of calibration is indicated on the online platform.

## 2.4 Output / Available data on the online platform

The online platform (accessible at https://simstrat.eawag.ch) is automatically fed every week with model results, metadata and plots for all the 54 modelled lakes (see Figure 2). It allows for efficient display and open sharing of the model results for interested users. While the framework is here restricted to Swiss lakes, the code could be easily adapted to other lakes outside Switzerland and used at the global scale. From the model results, we directly obtain time series of several model output variables. Those dataset include temperature, salinity, Brunt-Väisälä frequency, vertical diffusivity, and ice thickness. In
addition, we use the following known physical and lake-related properties: the acceleration of gravity $g = 9.81$ m$^2$ s$^{-1}$, the heat capacity of water $c_\mathrm{p} = 4.18 \cdot 10^3$ J K$^{-1}$ kg$^{-1}$, the volume of the lake $V$ [m$^3$], the area $A_z$ [m$^2$], temperature $T_z$ [°C], and density $\rho_z$ [kg m$^{-3}$] at depth $z$ [m], and the mean lake depth $\bar{z} = \frac{1}{V} \int z \, A_z \, dz$ [m] to calculate time series of derived values:

- Mean lake temperature: $\bar{T} = \frac{1}{V} \int T_z \, A_z \, dz$ [°C]
- Heat content: $H = c_\mathrm{p} \int \rho_z \, T_z \, A_z \, dz$ [J]
- Schmidt stability: $S_\mathrm{T} = \frac{g}{A_0} \int (z - \bar{z}) \, \rho_z \, A_z \, dz$ [J m$^{-2}$]
- Timing of summer stratification: we use a threshold based on the Schmidt stability to determine beginning and end of summer stratification. The lake is assumed to be stratified for $S_T / z_{lake} \geq 10$ J m$^{-3}$. Using a different criterion (e.g., temperature difference between surface and bottom water) results in variations in the calculated stratification period; however, the general pattern among lakes remains similar.
- Timing of ice cover: we use the existence of ice to determine beginning and end of ice covered period.

From these results, we create static and interactive plots. The latter are created using the Plotly Python Library (see https://plot.ly/python). The plots can be categorized as follows:

- History (e.g., contour plot of the whole temperature time series, line plot of the whole time series of Schmidt stability);
- Current situation (e.g., latest temperature profile);
- Statistics (e.g., average monthly temperature profiles, long-term trends).

All Output and processed data are directly available from the online platform.

## 3 Results and discussion

Analysis of model output allows to compare the response of the different systems to specific events or to long term changes. The Simstrat model web interface provides regional long-term high-frequency data updated in near real-time as output. This
represents a novel way to monitor, analyse and visualize processes in aquatic systems, and, most importantly, grant the entire community direct access to the findings. The coupling between Simstrat and PEST provides an effective way to calibrate

model parameters. The uncertainty quantification finally allows an appropriate informed use of the output data. Yet, more advanced methods for both parameter estimation and uncertainty quantification such as Bayesian inference (Gelman et al., 2013) should be applied to Simstrat.

Out of the 46 calibrated lakes, the post-calibration root mean square error (RMSE) is < 1 °C for 17 lakes, between 1 and 1.5 °C for 15 lakes, between 1.5 and 2 °C for 8 lakes and between 2 °C and 3°C for 6 lakes (Figure 3), calibration data was not sufficient for 8 lakes in which we used standard settings. This is comparable to the RMSE range of ~0.7-2.1 °C reported in a recent global 32-lake modelling study using GLM (Bruce et al., 2018) also including Lake Geneva, Lake Constance and Lake Zurich. The correlation coefficient remains always higher than 0.93 suggesting also that the model successfully reproduce the thermal structure of the investigated lakes Overall, the quality of the results is better for lowland lakes than for high altitude lakes where local meteorological and watershed information are often missing.

We illustrate the potential of high-frequency lake model data with two examples: first by briefly showing the long-term changes caused by climate change in Lake Brienz (section 3.1), and secondly by investigating the differential response of lakes across Switzerland to episodic forcing (short-term extremes, section 3.2).

**3.1 Long-term evolution of the thermal structure of lakes in response to climate trends**

Over the period 1981–2015, yearly averaged simulated surface temperatures in Lake Brienz increased with a significant (p<0.001) trend of +0.69 °C/decade (Figure 4a). For the same period, monthly in situ observations indicate a similar trend of 0.72 °C/decade (p~0.07), while the trend of air temperature at the meteorological station in Interlaken is lower (+0.50 °C/decade, p<0.01). Based on physical principles, lake surface temperature is expected to increase less than air temperature (Schmid et al., 2014), however Schmid and Köster (2016) also observed a higher trend in lake surface temperature than in air temperature for Lower Lake Zurich and assigned the excess warming to a positive trend in solar radiation. For the period 1981-2015, the trend in solar radiation is 5 W/m$^2$/decade that corresponds to an equilibrium temperature increase of about 0.2°C/decade. The warming rate at the surface of Lake Brienz is larger than observed trends in neighbouring lakes with reported increases of +0.46 °C/decade for Upper Lake Constance (1984 – 2011, Fink et al. 2014), +0.41°C/decade for Lower Lake Zürich (1981 - 2013, Schmid and Köster, 2016; 1955 - 2013, Livingstone, 2003), +0.55°C/decade for Lower Lake Lugano (1972 – 2013, Lepori and Roberts, 2015). This can be explained by the lower light penetration in Lake Brienz (ranging from ~1 m to ~10 m) compared to other light; the increase in solar radiation being distributed into a shallower layer and thereby warming slightly more the lake surface.

The temperature increase was significantly smaller in the hypolimnion, with a minimum trend at the lake bottom of 0.16 °C/decade (p<0.001), leading to a depth-averaged rate of temperature increase of 0.22 °C/decade (p<0.001). The temperature difference between the inflow and the outflow also contributes to the heat budget. While no significant change in the yearly total discharge was observed at the gauging stations of FOEN for the inflows Aare and Lütschine for the period 1981 – 2015,

the weighted inflow temperature increased by 0.26 °C/decade. The riverine temperature remains colder than the lake surface temperature leading to a yearly average loss of energy by through-flow of ~ - 40 W/m$^2$ for 2015. This result is consistent with the recent observations of Råman Vinnå (2018) suggesting that tributaries significantly affect the thermal response of lakes with residence time up to 2.7 years (as Lake Brienz). The contribution of the river to the heat budget of Lake Brienz is also ~

4 times larger than that previously estimated for Upper Lake Constance (Fink et al. 1994), a lake with a longer residence time. The increasing difference over time between the inflow temperature and the outflow temperature (taken as the lake surface temperature) leads to a non-negligible cooling contribution from the river of ~ 0.14 °C/decade (p<0.05). The temporal change in the discharge and its temperature resulting from climate change should therefore be taken into account in predicting the change in lake thermal structure.

The vertically heterogeneous warming modelled in Lake Brienz is consistent with previous observations showing that the difference in warming between the surface and the bottom increases the strength and duration of the stratified period (Zhong et al., 2016; Wahl and Peeters, 2014). We simulate an earlier onset of the stratification in spring of -7.5 day/decade (p<0.001) and a later breakdown of the stratification by +3.7 day/decade (p<0.001) (Figure 4c). Both the warming trend and the increase in length of the stratified period increase the Schmid stability (Figure 4d) and heat content (Figure 4f). Finally, the yearly

maximum stratification strength (Brunt-Väisälä frequency, Figure 4e) gradual increases over the investigated period with a rate of  3.3 x 10$^{-4}$ s$^{-2}$/decade. The simulated increase in overall stability (Figures 4d, 4e and 4f) reduces vertical mixing and affects the vertical storage of heat with less heat transferred immediately below the thermocline causing a slight decrease in temperature observed in autumn at ~30 m depth (Figure 4b). This effect is even more clearly seen in other lakes like Lake Geneva ( https://simstrat.eawag.ch/LakeGeneva) with the surface waters warming strongly (+1 °C/decade in June), resulting

in a cooling layer between 20 and 60 m (-0.2 °C/decade) in late summer. Such a reduction of vertical exchange is self-strengthening and enhances the differential vertical warming.

Such analyses can be extended to all modelled lakes. An inter-comparison of the temporal extent of summer stratification and winter ice cover period is illustrated in Figure 5. An altitude-dependent decrease of the duration of summer stratification is observed, along with a stronger corresponding increase in the duration of the inverse winter stratification from 1200 m. a.s.l.

This is possibly linked to an altitude dependency of climate-driven warming in Swiss lakes, first reported by Livingstone et al. (2005), which may be caused by a delay in meltwater runoff (Sadro et al., 2018). Here this process is not directly resolved but incorporated through the calibration procedure spanning all seasons.

In conclusion, the online platform provides all the data to estimate the past rate of warming of lakes and evaluate how the different external processes contribute to their heat budgets. The change in the thermal structure depends mostly to the change

in atmospheric forcing, yet, other factors such as the changes in discharge and temperature from the tributaries and the light absorption into the lake should also be taken into account. We specifically show that the rate of warming of the lake surface temperature significantly differs from that of depth-averaged temperature, thereby highlighting the benefit of using either *in-*

*situ* observations resolving the thermal structure over the water column or hydrodynamic model output for assessing climate change impacts on lake thermal structure.

## 3.2 Event based evolution of the lake thermal structure.

A major drawback of traditional lake monitoring programs in Switzerland is the coarse temporal resolution, with measurements often performed on a monthly basis. Thought sufficient for direct long-term trend studies as shown in section 3.1 when conducted over an extended period typically longer than 30 years (Gray et al., 2018). However, traditional monitoring programs cannot resolve the impact of short-term events and their consequences for the ecosystem. This is a strength of high-frequency (hourly time scale) lake modelling, which allows for simulation and comparison of the effects associated with rapid and often severe events such as storms. Based on high-frequency observations, Woolway et al. (2018) showed the effects of a major storm on Lake Windermere. They observed a decrease in the strength of the stratification, a deepening of the thermocline and the onset of internal waves oscillations ultimately upwelling oxygen depleted cold water into the downstream river. Furthermore, Perga et al. (2018) illustrated how storms could be just as important as gradual long-term trends for changes in light penetration and thermal structure in an Alpine lake.

Here we demonstrate how high-frequency model output can be used to study the influence of specific events on the thermal dynamics of lakes. As an example, we focus on the 28[th] of June 2018 when Switzerland experienced a strong but by no means exceptional storm with Northeasterly winds mainly affecting the North-Western part of the country – the mean wind speed during that day is shown spatially in Figure 6a. The evolution of the stratification strength, illustrated here by the Schmidt stability, is given in Figure 6b for one of the most affected lakes, Lake Neuchâtel ([https://simstrat.eawag.ch/LakeNeuchatel](https://simstrat.eawag.ch/LakeNeuchatel), Figure 2). This lake, with the main axis well-aligned to synoptical winds, experienced a ~8 % decrease in the Schmidt stability over this half-day event. Yet, the effects were not long-lasting and the Schmidt stability reverted to its pre-storm value within ~5 days (Figure 6b). This also resulted in a total increase of the lake heat content by ~$1.4 \cdot 10^{16}$ J from the start of the storm to the time of recovery. We used the Schmidt stability recovery duration as a way to assess the short-term effect of the storm on the different modelled lakes. In Figure 6a, lakes are coloured based on the delay in Schmidt stability increase (in days) caused by the storm. The impact of the storm was not limited to Lake Neuchâtel but rather showed a regionally-varying pattern. Particularly small- to medium-sized lakes in the North-Western parts of Switzerland were more affected than large lakes or lakes located in the Southern part of Switzerland. However, the thermal structure of these lakes quickly reverted to the seasonal early summer warming trend.

So far, climate-driven warming has been recognized to cause an overall increase in lake stratification strength and duration, and a gradual warming of the different layers (Schwefel et al., 2016; Zhong et al., 2016; Wahl and Peeters, 2014). Air temperature trend was the most studied forcing parameter. Yet, the dynamics of extreme events (such as heat waves, drought spells, storms), including their changes in strength and distribution, has been comparatively overlooked. Scenario exploration,

climate change studies, or historical forcing reanalysis should be integrated in such web-based hydrodynamic platforms to assess their roles in modifying the lake thermal structures and heat storage.

**4 Conclusion**

The workflow presented in this paper allows open sharing of high-frequency, up-to-date and permanently available lake model results for multiple users and purposes. We demonstrated the benefit of the platform through two simple case studies. First, we showed that the high frequency modelled temperature data allows a complete assessment of the effect of climate change on the thermal structure of a lake. We specifically show the need to evaluate changes in all atmospheric forcing, in the watershed or through-flow heat energy and in light penetration to accurately assess the evolution of the lake thermal structure. Then we showed that the high frequency modelled data can be used to investigate special events such as wind storms, there in-situ measurements under current temporal resolution are failing. More generally, these results are well suited for the following applications and target groups:

- For the public, the platform serves as an informative website enabling easy access to broad quantities of regional scientific results, with the intention of raising interest about lake ecosystem dynamics.
- For lake managers, the platform makes relevant information available, such as (i) near real time temperature and stratification conditions of the lakes, (ii) simple statistical analyses such as monthly temperature profiles and long-term temperature trends.
- For researchers, this work can facilitate (i) scenario modelling of any of the lakes, as the basic model setup is ready-to-use, (ii) improvement of the lake model with addition of previously unresolved processes (e.g., resuspension with changed light properties), (iii) access to variables that were previously not or irregularly available (e.g., vertical diffusivity, heat content, stratification and heat fluxes), and (iv) specific comparative analyses, whereby a given question can be investigated simultaneously over many lakes (e.g., the impact of climate change or a regional storm).

By promoting a cross-exchange of expertise through openly sharing of in-situ and model data at high frequency, this open-access data platform is a new path forward for scientists and practitioners.

**Code and data availability**

The workflow was developed for Swiss lakes but can be easily extended to other geographical area or at global scale by using other meteorological input data. Simstrat and the Python workflow are available on https://github.com/Eawag-AppliedSystemAnalysis/Simstrat/releases/tag/v2.1 (https://doi.org/10.5281/zenodo.2600709) and https://github.com/Eawag-AppliedSystemAnalysis/Simstrat-WorkflowModellingSwissLakes (https://doi.org/10.5281/zenodo.2607153). Meteorological

data are available from MeteoSwiss (https://gate.meteoswiss.ch/idaweb/), hydrological data are available from FOEN (https://www.hydrodaten.admin.ch), CTD data were provided by various sources listed here: https://simstrat.eawag.ch/impressum . The calibration software PEST is available on http://www.pesthomepage.org/

## Author contribution

The new version of Simstrat was developed by FB, AG and LRV. The workflow was developed by AG. The ice model was developed by LVR. The concept of the workflow was defined by DB. All authors contributed to the validation of the model and interpretation of the results. AG and DB wrote the manuscript with contributions from FB, LVR and MS.

## Competing interests

The authors declare that they have no conflict of interest

## Acknowledgment

We thank Davide Vanzo for helping with the Docker and the scripts and Michael Pantic for helping restructuring the version 2.1 of Simstrat. We finally thank Marie-Elodie Perga for her comments on a preliminary version of the manuscript. The full list of acknowledgements regarding in-situ observations can be found here: https://simstrat.eawag.ch/impressum .

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

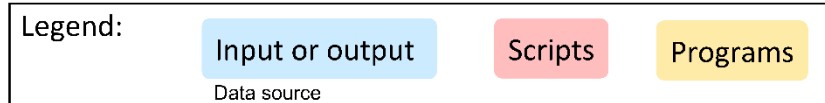

Figure 1. General workflow diagram. Model input (left box) is retrieved and processed by the Python script "Simstrat.py", which runs the model (Simstrat v2.1) and/or model calibration (using PEST v15.0) and produces output (right box). This output is then uploaded to a web interface (https://simstrat.eawag.ch) for general use. All scripts and programs are available on https://github.com/Eawag-AppliedSystemAnalysis/Simstrat/releases/tag/v2.1 and https://github.com/Eawag-AppliedSystemAnalysis/Simstrat-WorkflowModellingSwissLakes. Simstrat = one dimensional hydrodynamic model; CTD = Conductivity, Temperature, Depth profiler; PEST = Model independent parameter estimation and uncertainty analysis software; FOEN = Swiss Federal Office of Environment; MeteoSwiss = Swiss Federal Office of Meteorology and Climatology; Swisstopo = Swiss Federal Office of Topography

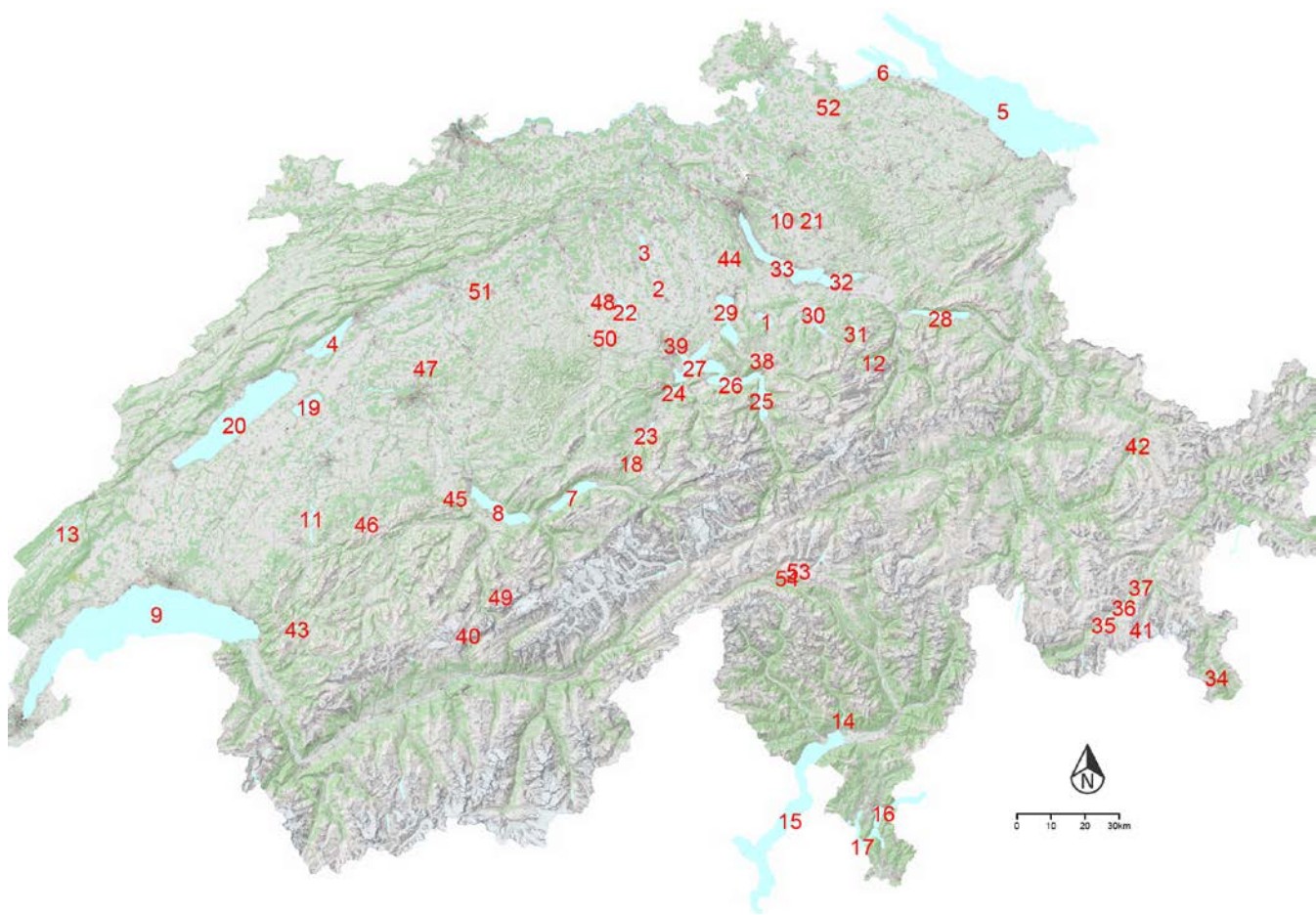

5    **Figure 2. Illustration of the interactive map displayed on the homepage of the online platform: https://simstrat.eawag.ch. The location of the lakes discussed in this manuscript is also indicated with numbers (See Appendix A). Basemap is provided by Swisstopo and the specific legend can be found here https://api3.geo.admin.ch/static/images/legends/ch.swisstopo.swisstlm3d-karte-farbe_en_big.pdf**

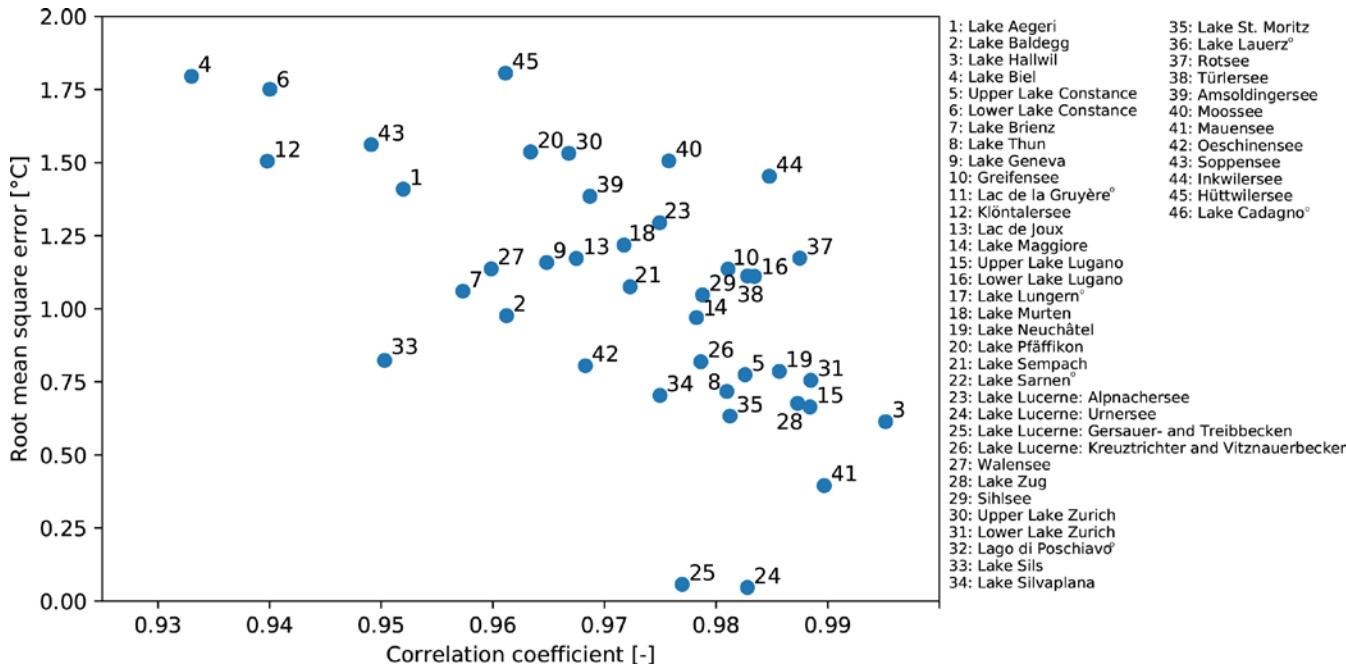

**Figure 3. Performance of the model for the different lakes, as shown by the root mean square error (RMSE) and the correlation coefficient. Six lakes (with symbol º on the legend) with RMSE > 2 °C are not shown.**

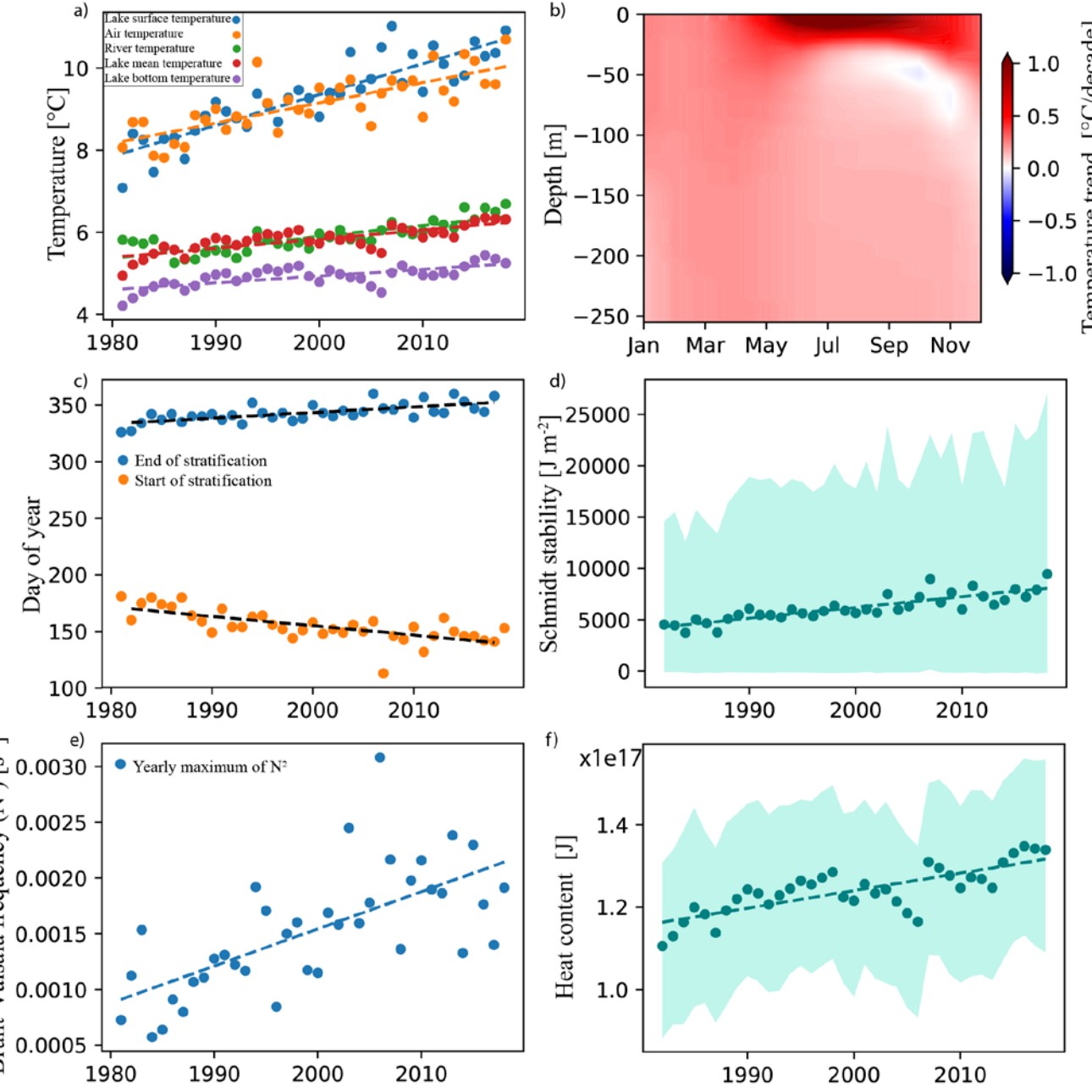

**Figure 4. Evolution of several indicators for Lake Brienz over the period 1981-2018; all linear regression have p_values << 0.001:** (a) yearly mean lake surface temperature (0.69 °C/decade), yearly mean air temperatures (0.49 °C/decade), yearly mean tributary temperatures (0.26 °C/decade), yearly mean lake temperatures (0.22 °C/decade) and yearly mean bottom temperatures (0.16 °C/decade), with linear regression, (b) contour plot of the linear temperature trend through depth and month, (c) yearly start (+3.7 days/decade) and end (-7.5 days/decade) day of summer stratification, with linear regression, (d) yearly mean (line), min and max (shaded area) Schmidt stability, with linear regression, (e) yearly maximum Brunt-Väisälä frequency ($3.3 \times 10^{-4}$ $1/s^2$/decade), with linear regression (f) yearly mean (line), min and max (shaded area) heat content.

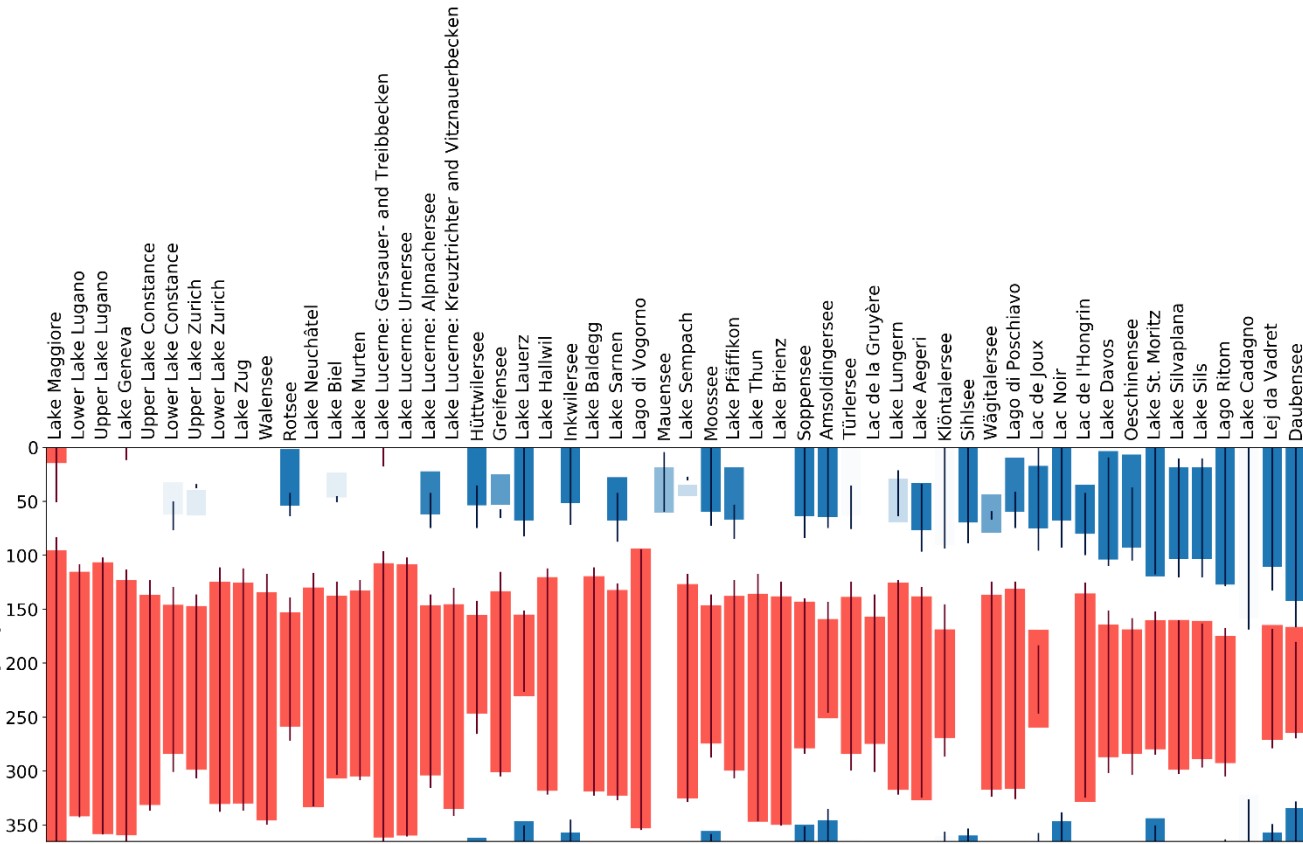

**Figure 5. Comparison of timing of stratification and ice cover for the considered lakes. The coloured areas represent the mean periods of summer stratification (red) and ice cover (blue); the vertical lines represent the last year (here, 2017). The transparency for the ice cover indicates the freezing frequency: full transparency means that ice was never modelled, while no transparency means that ice was modelled every winter. Lakes are ordered from left (low elevation) to right (high elevation). The time period of data used is indicated in Appendix A.**

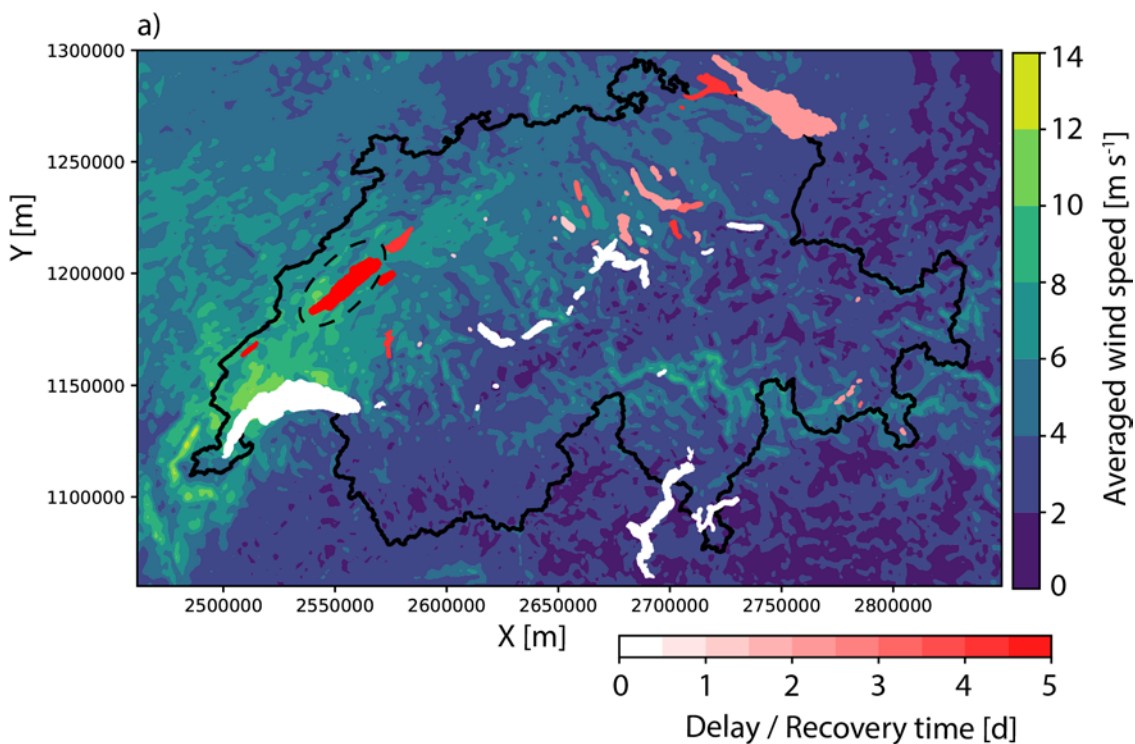

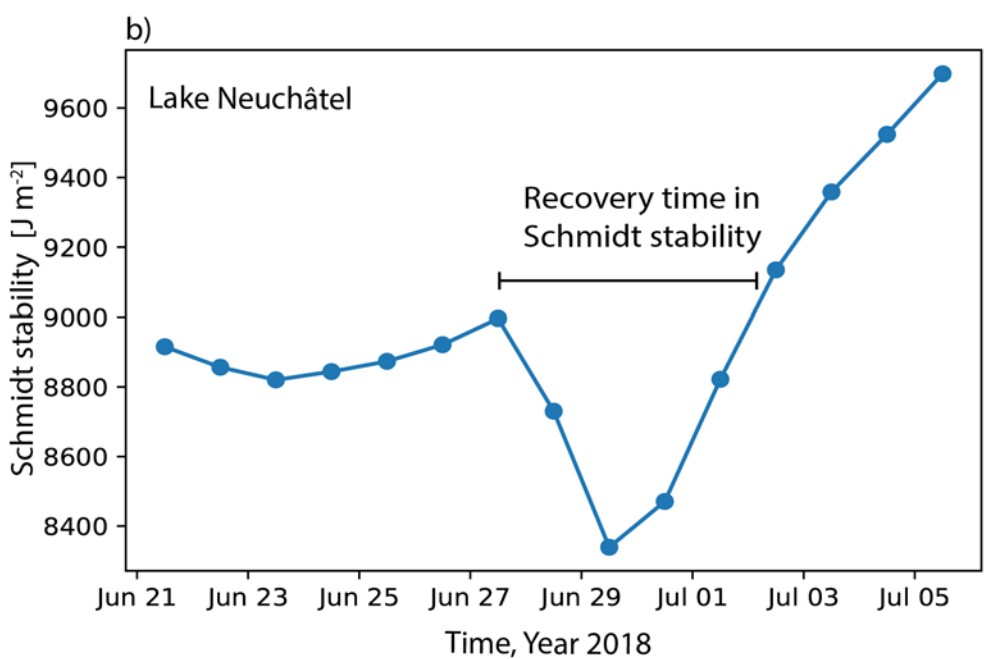

**Figure 6. (a) Mean wind field on June 28[th], 2018 (data source: MeteoSwiss, COSMO-1 model, coordinate system CH1903+) and delay in Schmidt stability increase for the modelled lakes: from no delay (white) to a delay of more than 5 days (red). (b) Schmidt stability (daily average) in Lake Neuchâtel during the period of the storm.**

**Table 1. Input data sources used for the model**

| Data | Source | Model input |
|---|---|---|
| Lake bathymetry | Swisstopo (https://www.swisstopo.admin.ch) | Bathymetry profile |
| Meteorological forcing | MeteoSwiss (http://meteoswiss.admin.ch) | Air temperature, solar radiation, humidity, wind, cloud cover, precipitation |
| Hydrological forcing | FOEN (http://hydrodaten.admin.ch) | Inflow discharge, inflow temperature |
| Secchi depth | Eawag, cantonal monitoring | Light absorption coefficient |
| CTD profiles | Eawag, cantonal monitoring | Initial conditions, temperature observations for calibration |

**Table 2. Model parameters. The asterisk (\*) indicates the parameters that were calibrated. The geothermal heat flux is based on existing geothermal data for Switzerland: https://www.geocat.ch/geonetwork/srv/eng/md.viewer#/full_view/2d8174b2-8c4a-44ea-b470-cb3f216b90d1.**

| Parameter | Description and units | Default value |
| --- | --- | --- |
| lat | Latitude [°] | Based on lake location |
| p_air | Air pressure [mbar] | Based on lake elevation |
| a_seiche* | Ratio of wind energy going into seiche energy [-] | Based on lake size |
| q_nn | Fractionation coefficient for seiche energy [-] | 1.10 |
| f_wind* | Scaling factor for wind speed [-] | 1.00 |
| c10 | Scaling factor for the wind drag coefficient [-] | 1.00 |
| cd | Bottom drag coefficient [-] | 0.002 |
| hgeo | Geothermal heat flux [$W/m^2$] | Based on geothermal map (see table caption) |
| p_radin* | Scaling factor for the incoming long wave radiation [-] | 1.00 |
| p_windf | Scaling factor for the fluxes of sensible and latent heat [-] | 1.00 |
| albsw | Albedo of water for short wave radiation [-] | 0.09 |
| beta_sol | Fraction of short wave radiation absorbed as heat in the uppermost water layer [-] | 0.35 |
| p_albedo* | Scaling factor for snow/ice albedo, thereby affecting melting and under ice warming [-] | 1.00 |
| freez_temp | Water freezing temperature [°C] | 0.01 |
| snow_temp | Temperature below which precipitation falls as snow [°C] | 2.00 |

**Appendix**

### A. Properties of the modelled lakes

The following table summarizes the main properties of the 54 lakes we model in this work. The full dataset is available as a JSON file. An asterisk after the lake name indicates that this lake was not calibrated due to the lack of observational data. MeteoSwiss = (Swiss) Federal Office of Meteorology and Climatology. FOEN = (Swiss) Federal Office for the Environment. # indicates lakes where secchi disk depth are available. For lakes with clearly defined multi basin such as Lake Lucerne, Lake Zurich, Lake Constance and Lake Lugano, each basin is considered as a separated lake connected to the other basins by inflows/outflows

| Lake | | Volume [km³] | Surface [km²] | Max depth [m] | Retention time [y] | Elevation [m] | Trophic state | Weather station IDs (MeteoSwiss) | Hydrological station IDs (FOEN) | Model timeframe |
|---|---|---|---|---|---|---|---|---|---|---|
| Lake Aegeri 689574 / 191747 | 1 | 0.36 | 7.3 | 83 | ~ 6.8 | 724 | O | AEG, SAG, EIN | - | 2012-2018 |
| Lake Baldegg 662239 / 228077 | 2 | 0.174 | 5.2 | 66 | ~ 4.2 | 463 | E | MOA | - | 2012-2018 |
| Lake Hallwil 658779 / 237484 | 3 | 0.285 | 10.3 | 48 | ~ 3.9 | 449 | E | MOA | 2416 | 2012-2018 |
| Lake Biel 578599 / 214194 | 4 | 1.12 | 39.3 | 74 | ~ 0.16 | 429 | E# | CRM | 2085, 2307, 2446 | 1993-2018 |
| Upper Lake Constance 749649 / 275225 | 5 | 47.6 | 473 | 251 | ~ 4.3 | 395 | M# | ARH, GUT | 2473, 2308, 2312 | 1981-2018 |
| Lower Lake Constance 718479 / 285390 | 6 | 0.8 | 63 | 45 | ~ 0.05 | 395 | M | STK, HAI, GUT | - | 1981-2018 |

| | | | | | | | | | | |
|---|---|---|---|---|---|---|---|---|---|---|
| Lake Brienz<br>640709 / 175275 | 7 | 5.17 | 29.8 | 259 | ~ 2.7 | 564 | O# | INT | 2019, 2109 | 1981-2018 |
| Lake Thun<br>619899 / 172630 | 8 | 6.5 | 48.3 | 217 | ~ 1.9 | 558 | O# | THU, INT | 2457, 2469, 2488 | 1981-2018 |
| Lake Geneva<br>533600 / 144624 | 9 | 89 | 580 | 309 | ~ 11 | 372 | M# | PUY | 2009, 2432, 2433, 2486, 2493 | 1981-2018 |
| Greifensee<br>693699 / 245032 | 10 | 0.15 | 8.5 | 32 | ~ 1.1 | 435 | E | SMA | - | 1981-2018 |
| Lac de la Gruyère<br>573990 / 168654 | 11 | 0.22 | 9.6 | 75 | ~ 0.4 | 677 | NA | MAS, GRA | 2160, 2412 | 2011-2018 |
| Klöntalersee<br>716984 / 209627 | 12 | 0.056 | 3.3 | 45 | ~ 0.5 | 848 | O | GLA | - | 1981-2018 |
| Lac de Joux<br>511590 / 165965 | 13 | 0.145 | 8.77 | 32 | 0.85 | 1004 | M | CHB, BIE | - | 2009-2018 |
| Lago di Vogorno*<br>709279 / 118833 | 14 | 0.1 | 1.68 | 204 | A | 470 | O | OTL | 2605 | 1981-2018 |
| Lake Maggiore<br>694300 / 92576 | 15 | 37 | 212 | 372 | ~ 4 | 193 | O | OTL | 2068, 2368 | 1981-2018 |

| Upper Lake Lugano 721139 / 95471 | 16 | 4.69 | 27.5 | 288 | ~12.3 | 271 | E | LUG | 2321 | 1981-2018 |
| Lower Lake Lugano 714239 / 86391 | 17 | 1.14 | 20.3 | 95 | ~ 1.4 | 271 | M | LUG | 2629, 2461 | 1981-2018 |
| Lake Lungern 655099 / 183325 | 18 | 0.065 | 2 | 68 | ~ 0.6 | 688 | NA | GIH | - | 2010-2018 |
| Lake Murten 572700 / 198094 | 19 | 0.55 | 22.8 | 45 | ~ 1.2 | 429 | M# | NEU | 2034 | 1981-2018 |
| Lake Neuchâtel 554800 / 194974 | 20 | 13.8 | 218 | 152 | ~ 8.2 | 429 | M# | NEU | 2378, 2369, 2480, 2458, 2447 | 1981-2018 |
| Lake Pfäffikon 701604 / 245377 | 21 | 0.059 | 3.3 | 36 | ~ 2.1 | 537 | M | SMA | - | 1981-2018 |
| Lake Sempach 654629 / 221355 | 22 | 0.66 | 14.5 | 87 | ~ 16.9 | 504 | M | EGO | 2608 | 2010-2018 |
| Lake Sarnen 658349 / 190767 | 23 | 0.239 | 7.5 | 51 | ~ 0.8 | 469 | O | GIH | - | 2010-2018 |
| Lake Lucerne: Alpnachersee 667144 / 202267 | 24 | 0.1 | 4.5 | 35 | ~ 0.3 | 434 | O | LUZ | 2102, 2436 | 1981-2018 |

| | | | | | | | | | | |
|---|---|---|---|---|---|---|---|---|---|---|
| Lake Lucerne: Urnersee 688649 / 200895 | 25 | 3.16 | 22 | 200 | ~ 2.0 | 434 | O | ALT | 2056, 2276 | 1981-2018 |
| Lake Lucerne: Gersauer- and Treibbecken 681659 / 203585 | 26 | 4.41 | 30 | 214 | ~ 1.6 | 434 | O | GES, ALT | 2084, 2481 | 1981-2018 |
| Lake Lucerne: Kreuztrichter and Vitznauerbecken 672049 / 208875 | 27 | 4.35 | 59 | 151 | ~ 0.7 | 434 | O | LUZ | - | 1981-2018 |
| Walensee 735739 / 202690 | 28 | 2.5 | 24.2 | 151 | ~ 1.4 | 419 | O | QUI, LAC, GLA | 2372, 2426 | 1981-2018 |
| Lake Zug 680049 / 216865 | 29 | 3.2 | 38.3 | 197 | ~ 14.7 | 417 | E | CHZ, WAE | 2477 | 1981-2018 |
| Sihlsee 701504 / 222387 | 30 | 0.096 | 11.3 | 22 | ~ 0.4 | 889 | O | EIN | 2300, 2635 | 2012-2018 |
| Wägitalersee* 701504 / 222387 | 31 | 0.15 | 4.18 | 65 | ~ 1.6 | 900 | O | LAC, EIN | - | 2012-2018 |
| Upper Lake Zurich 707159 / 229595 | 32 | 0.47 | 20.3 | 48 | ~ 0.69 | 406 | M | WAE | 2104 | 1981-2018 |

| Lower Lake Zurich 687209 / 237715 | 33 | 3.36 | 68.2 | 136 | ~ 1.4 | 406 | M# | LAC, SCM, WAE | - | 1981-2018 |
|---|---|---|---|---|---|---|---|---|---|---|
| Lago di Poschiavo 804706 / 128871 | 34 | 0.12 | 1.98 | 85 | ~ 0.5 | 962 | O | ROB | 2078 | 1981-2018 |
| Lake Sils 776533 / 143922 | 35 | 0.137 | 4.1 | 71 | ~ 2.2 | 1797 | O | SIA | - | 2014-2018 |
| Lake Silvaplana 780801 / 146926 | 36 | 0.14 | 2.7 | 77 | ~ 0.7 | 1791 | O | SIA | - | 2014-2018 |
| Lake St. Moritz 784870 / 152099 | 37 | 0.02 | 0.78 | 44 | ~ 0.1 | 1768 | O | SAM | 2105 | 1981-2018 |
| Lake Lauerz 688864 / 209546 | 38 | 0.0234 | 3.07 | 14 | ~ 0.3 | 447 | M | GES, LUZ | - | 1981-2018 |
| Rotsee 666491 / 213558 | 39 | 0.00381 | 0.48 | 16 | ~ 0.4 | 419 | E | LUZ | - | 1981-2018 |
| Daubensee* 613862 / 140026 | 40 | | 0.64 | 50 | NA | 2207 | O | BLA | - | 2013-2018 |
| Lej da Vadret* 785308 / 141515 | 41 | | 0.43 | 50 | NA | 2160 | O | SIA | - | 2014-2018 |

| | | | | | | | | | | |
|---|---|---|---|---|---|---|---|---|---|---|
| Lake Davos*<br><br>784261 /<br><br>188317 | 42 | 0.0156 | 0.59 | 54 | NA | 1558 | O | DAV | - | 1981-2018 |
| Lac de<br>l'Hongrin*<br><br>569975 /<br><br>141537 | 43 | 0.0532 | 1.6 | 105 | NA | 1250 | O | CHD | - | 2012-2018 |
| Türlersee<br><br>680514 /<br><br>235858 | 44 | 0.00649 | 0.497 | 22 | ~ 2 | 643 | E | WAE | - | 1981-2018 |
| Amsoldingersee<br><br>610534 /<br><br>174906 | 45 | 0.00255 | 0.382 | 14 | NA | 641 | E# | THU | - | 2012-2018 |
| Lac Noir*<br><br>587970 /<br><br>168280 | 46 | 0.00252 | 0.47 | 10 | NA | 1045 | M | PLF | - | 1989-2018 |
| Moossee<br><br>603165 /<br><br>207928 | 47 | 0.00339 | 0.31 | 21 | NA | 521 | E | BER | - | 1981-2018 |
| Mauensee<br><br>648258 /<br><br>224587 | 48 | | 0.55 | 9 | NA | 504 | E | EGO | - | 2010-2018 |
| Oeschinensee<br><br>622116 /<br><br>149701 | 49 | 0.0402 | 1.11 | 56 | ~ 1.6 | 1578 | O | ABO | - | 1983-2018 |
| Soppensee<br><br>648765 /<br><br>215720 | 50 | 0.00286 | 0.25 | 27 | ~ 3.1 | 596 | E | EGO | - | 2010-2018 |

| | | | | | | | | | | |
|---|---|---|---|---|---|---|---|---|---|---|
| Inkwilersee<br><br>617009 /<br>227527 | 51 | 0.00094 | 0.102 | 6 | ~ 0.1 | 461 | E# | KOP | - | 2011-2018 |
| Hüttwilersee<br><br>705538 /<br>274275 | 52 | | 0.34 | 28 | NA | 434 | E | HAI | - | 2010-2018 |
| Lake Cadagno<br><br>697683 /<br>156223 | 53 | 0.00242 | 0.26 | 21 | ~ 1.5 | 1921 | E | PIO | - | 1981-2018 |
| Lago Ritom*<br><br>695933 /<br>155169 | 54 | 0.048 | 1.49 | 69 | NA | 1850 | O | PIO | - | 1981-2018 |

| Meteorological station | Abreviation | Altitude (m.a.s.l) | Coordinates (CH) |
|---|---|---|---|
| Oberägeri | AEG | 724 | 688728 / 220956 |
| Sattel | SAG | 790 | 690999 / 215145 |
| Einsiedeln | EIN | 911 | 699983 / 221068 |
| Mosen | MOA | 453 | 660128 / 232851 |
| Cressier | CRM | 430 | 571163 / 210797 |
| Altenrhein | ARH | 398 | 760382 / 261387 |
| Güttingen | GUT | 440 | 738422 / 273963 |
| Steckborn | STK | 397 | 715871 / 280916 |
| Salen-Reutenen | HAI | 719 | 719099 / 279047 |
| Interlaken | INT | 577 | 633023 / 169092 |
| Thun | THU | 570 | 611201 / 177640 |
| Pully | PUY | 456 | 540819 / 151510 |
| Zürich / Fluntern | SMA | 556 | 685117 / 248066 |
| Marsens | MAS | 715 | 571758 / 167317 |
| Fribourg / Posieux | GRA | 651 | 575184 / 180076 |
| Les Charbonnières | CHB | 1045 | 513821 / 169387 |
| Bière | BIE | 684 | 515888 / 153210 |
| Locarno / Monti | OTL | 367 | 704172 / 114342 |
| Lugano | LUG | 273 | 717874 / 95884 |
| Giswil | GIH | 471 | 657322 / 188976 |
| Neuchâtel | NEU | 485 | 563087 / 205560 |
| Egolzwil | EGO | 522 | 642913 / 225541 |

| | | | |
|---|---|---|---|
| Luzern | LUZ | 454 | 665544 / 209850 |
| Altdorf | ALT | 438 | 690180 / 193564 |
| Gersau | GES | 521 | 682510 / 205572 |
| Quinten | QUI | 419 | 734848 / 221278 |
| Laschen / Galgenen | LAC | 468 | 707637 / 226334 |
| Glarus | GLA | 517 | 723756 / 210568 |
| Cham | CHZ | 443 | 677758 / 226878 |
| Wädenswil | WAE | 485 | 693847 / 230744 |
| Schmerikon | SCM | 408 | 713725 / 231533 |
| Plaffeien | PLF | 1042 | 586825 / 177407 |
| Segl-Maria | SIA | 1804 | 778575 / 144977 |
| Blatten, Lötschental | BLA | 1538 | 629564 / 141084 |
| Adelboden | ABO | 1322 | 609350 / 149001 |
| Piotta | PIO | 990 | 695880 / 152265 |

### B. Ice module

The ice and snow module employed is based on the work of Leppäranta (2014, 2010) and Saloranta and Andersen (2007), and includes the following physical processes:

- Air temperature dependent formation and growth of black ice, including the insulating effect of a snow cover.
- Snow layer build-up, including the compression effect due to the weight of fresh snow.
- Buoyancy-driven formation of white ice.
- Short wave irradiance reflection and penetration into the underlying water column.
- Melting of snow, white and black ice due to both the direct heat flux through the atmospheric interface and the absorption of short wave irradiance.

Three layers are used to represent black ice, white ice, and snow. An instant supply of water through cracks in the black ice is assumed to occur in order to form white ice. The water stored in ice and snow is neither withdrawn during ice formation nor added during melting to the water balance. Furthermore, the effect of liquid water pools on top of or between the layers is neglected

**Below the freezing point (ice formation)**

The ice module is activated as the water temperature in the topmost grid cell $T_w$ (°C) drops below the freezing temperature $T_f$ (°C). $T_f$ can be set to zero for a vertical grid size $\leq 0.5$ m, the user can adapt (raise) this value to fit coarser grids. If temperature is below the freezing point, the energy incorporated into the change of state $E_f$ is calculated as

$$E_f = \rho_w c_{pw} z_1 (T_f - T_w) \qquad \text{(B1)}$$

here $\rho_w$ (1000 kg m$^{-3}$) is the density of fresh water, $c_{pw}$ the heat capacity of water (4182 J kg$^{-1}$ °C$^{-1}$) and $z_1$ the height of the topmost grid cell. $E_f$ and the latent heat of freezing $l_h$ (3.34·10$^5$ J kg$^{-1}$) as well as the density of black ice $\rho_{ib}$ (916.2 kg m$^{-3}$) are used for calculating the initial height of black ice $h_{ib}$ (m) in Eq. B2, thereafter $T_w$ is set equal to $T_f$

$$h_{ib} = E_f / (l_h \rho_{ib}) \qquad \text{(B2)}$$

If an ice cover is present and if the atmospheric temperature $T_a$ (°C) is smaller or equal to $T_f$, the growth of black ice $dh_{ib}/dt$ continues as described in (Saloranta and Andersen, 2007).

$$\frac{dh_{ib}}{dt} = \sqrt{2k_i / (\rho_{ib} * l_h) * (T_f - T_i)} \qquad \text{(B3)}$$

Here $k_i$ (2.22 W K$^{-1}$ m$^{-1}$) is the thermal conductivity of ice at 0 °C and $T_i$ (°C) the ice temperature calculated as

$$T_i = \frac{P T_f + T_a}{1 + P} \qquad \text{(B4)}$$

$$P = \max\left( \frac{k_i h_s}{k_s h_{ib}}, \frac{1}{10 h_{ib}} \right) \qquad \text{(B5)}$$

There $k_s$ (0.2 W K$^{-1}$ m$^{-1}$) is the thermal conductivity of snow and $h_s$ (m) the height of the snow layer. When $T_a$ is smaller than the snow temperature (default set to 2 °C) water equivalent precipitation $p_r$ (m hour$^{-1}$) is turned into fresh snow $h_{s\_new}$ (m) as

$$h_{s\_new} = p_r \frac{\rho_w}{\rho_{s0}} \qquad \text{(B6)}$$

where $\rho_{s0}$ (250 kg m$^{-3}$) is the initial snow density. The existing snow cover $h_s$ (m) undergoes compression (first terms Eq. B7 and B8) by the new layer as described in Yen (1981), thereafter the new and existing layers are combined in both height and density (second terms of Eqs. B7 and B8).

$$\frac{dh_s}{dt} = -h_s \left( 1 - \frac{\rho_s}{[\rho_s + d\rho_s]} \right) + h_{s\_new}$$

(B7)

$$\frac{d\rho_s}{dt} = \rho_s C_1 w_s e^{-C_2 \rho_s} - \left( \rho_s - \frac{\rho_s h_s + \rho_{s0} h_{s\_new}}{h_s + h_{s\_new}} \right)$$

(B8)

here $\rho_s$ (kg m$^{-3}$) is the snow layer density kept within $\rho_{s0} < \rho_s < \rho_{sm}$ with the maximum snow density set to 450 kg m$^{-3}$, $C_1$ (5.8 m$^{-1}$ hour$^{-1}$) and $C_2$ (0.021 m$^3$ kg$^{-1}$) are snow compression constants, and $w_s$ (m) is the total weight above the layer under

5  compression expressed in water equivalent height.

If the snow mass $m_s$ (kg m$^{-2}$) becomes heavier than the upward acting buoyancy force $B_i$ (kg m$^{-2}$), white ice with height $h_{iw}$ (m) and density $\rho_{iw}$ (875 kg m$^{-3}$ Saloranta, 2000) is formed between the snow and the black ice layers to achieve equilibrium between $B_i$ and $m_s$.

$$B_i = h_{ib} \left( \rho_w - \rho_{ib} \right) + h_{iw} \left( \rho_w - \rho_{iw} \right)$$

(B9)

$$\frac{dh_{iw}}{dt} = \frac{m_s - B_i}{\rho_s}$$

(B10)

In this model, we assume continuous supply of water through cracks in the black ice to form white ice. The formation of white ice takes place instantaneously each time step and we do not consider the influence of pools under the snow for melting or short wave irradiance penetration.

**Above the freezing point (melting)**

15  If an ice cover is present and if $T_a > T_f$ melting starts. Each layer melts from above through the atmospheric interface and by penetrating short wave radiation

$$\frac{dh_{x\_upper}}{dt} = -\frac{H_{x\_y}}{\left( l_h + l_e \right) \rho_x}$$

(B11)

where $H_{x\_y}$ (W m$^{-2}$) is the layer-dependent heat flux (in the following, $_x$ represents $_s$, $_{iw}$ or $_{ib}$). The model supports melting through both sublimation (solid to gas) and non-sublimation (solid to liquid) with the inclusion/exclusion of the latent heat of

20  evaporation $l_e$ (J kg$^{-1}$). Non-sublimation melting is default with $l_e$ set to zero, for sublimation melting the user can set $l_e$ to 2265 kJ kg$^{-1}$. For the uppermost layer ($_y = {}_{top}$, Eq. B12) the heat flux includes layer dependent uptake of short wave radiation $H_{s\_x}$,

long wave absorption $H_a$ or layer dependent emission $H_{w\_x}$ as well as sensible $H_k$ and latent $H_v$ heat. If the layer is not in direct contact with the atmosphere, only $H_s$ is used for melting from above ($_{y\,=\,under}$, Eq. B13).

$$H_{x\_top} = H_{s\_x} + H_a + H_{w\_x} + H_k + H_v \tag{B12}$$

$$H_{x\_under} = H_{s\_x} \tag{B13}$$

5   Here we follow Leppäranta (2014, 2010) for determining the heat flux terms in Eq. B12. The transmittance of short wave irradiance through each layer depends on each layers thickness $h_x$ as well as on the layer specific bulk attenuation coefficient $\lambda_x$ (m$^{-1}$; default $\lambda_s = 24$, $\lambda_{iw} = 3$ and $\lambda_{ib} = 2$; Leppäranta, 2014).

$$H_{s\_s} = I_s A_p \left(1 - A_x\right)\left(1 - e^{\left(-\lambda_s h_s\right)}\right) \tag{B14}$$

$$H_{s\_iw} = I_s A_p \left(1 - A_x\right)\left(e^{\left(-\lambda_s h_s\right)} - e^{\left(-\lambda_s h_s - \lambda_{iw} h_{iw}\right)}\right) \tag{B15}$$

$$H_{s\_ib} = I_s A_p \left(1 - A_x\right)\left(e^{\left(-\lambda_s h_s - \lambda_{iw} h_{iw}\right)} - e^{\left(-\lambda_s h_s - \lambda_{iw} h_{iw} - \lambda_{ib} h_{ib}\right)}\right) \tag{B16}$$

$$H_{s\_w} = I_s A_p \left(1 - A_x\right)\left(1 - e^{\left(-\lambda_s h_s - \lambda_{iw} h_{iw} - \lambda_{ib} h_{ib}\right)}\right) \tag{B17}$$

There $H_{s\_w}$ is the radiation penetrating through the ice cover to the water below and $I_s$ (W m$^{-2}$) the incoming short wave irradiance. We introduce the albedo parameter $A_p$ which tunes short wave irradiance in order to match observed water temperatures, thus adjusting the melting and indirectly the duration of the ice cover. Furthermore, depending on which layer

15  is in contact with the atmosphere we use a layer dependent constant albedo $A_x$ (default $A_s = 0.7$, $A_{iw} = 0.4$ and $A_{ib} = 0.3$; Leppäranta, 2014).

$$A_x \begin{cases} A_s, h_s > 0 \\ A_{iw}, h_s = 0\,\&\,h_{iw} > 0 \\ A_{ib}, h_s + h_{iw} = 0 \end{cases} \tag{B18}$$

Calculating $H_a$ requires the long wave emission parameters $k_a = 0.68$, $k_b = 0.036$ (mbar$^{-1}$) and $k_c = 0.18$ (Leppäranta, 2010), atmospheric water vapour pressure $e_a$ (mbar), cloud cover C and Stefan Boltzmann's constant $\sigma$ (5.67*10$^{-8}$ W m$^{-2}$ K$^{-4}$). For

20  Eqs. B19 and B20 the temperature $T_x$ is given in Kelvin. $H_{w\_x}$ is layer dependent for the emissivity $E_x$ with $E_{iw} = E_{ib} = 0.97$ and $E_s(\rho_s)$ from 0.8 at $\rho_s = 250$ kg m$^{-3}$ to $E_s = 0.9$ for $\rho_s = 450$ kg m$^{-3}$. Calculating $H_k$ and $H_v$ requires the atmospheric density $\rho_{a\,=}$

1.2 kg m$^{-3}$, the heat capacity of air $c_{pa} = 1005$ J kg$^{-1}$ K$^{-1}$, the wind speed at 10 m height $w_{10}$, the convective ($b_c$) and latent ($b_l$) bulk exchange coefficients both set to 0.0015 (Leppäranta, 2010; Gill, 1982), as well as the specific humidity both measured $q_a$ (mbar) and at saturation $q_0$. There $q_a = 0.622e_a/p_a$ where $p_a$ is the air pressure and $q_0 = 0.622*6.11/p_a$ at $T_a = 0$ °C (Leppäranta, 2014).

$$H_a = \left(k_a + k_b\sqrt{e_a}\left[1 + k_c C^2\right]\right)\sigma T_a^4$$

(B19)

$$H_{w\_x} = E_x \sigma T_f^4$$

(B20)

$$H_k = \rho_a c_{pa} b_c \left(T_a - T_f\right) w_{10}$$

(B21)

$$H_v = \rho_a l_h b_l \left(q_a - q_0\right) w_{10}$$

(B22)

As $H_{s\_w}$ warms the water under the ice, melting takes place from underneath with the energy $H_{bottom}$ (W m$^{-2}$).

$$H_{bottom} = \left(T_w - T_f\right) c_{pw} \rho_w z_1$$

(B23)

After obtaining $H_{bottom}$ the temperature of the first cell is set to $T_f$ and the decrease of ice cover from below becomes

$$\frac{dh_{x\_lower}}{dt} = -\frac{H_{bottom}}{l_m \rho_x}$$

(B24)

Eq. B24 is only applied to $h_{ib}$ and $h_{iw}$. In principle, $h_s$ melts completely from above using Eq. B11 before $h_{ib}$ and $h_{iw}$ reach zero, however, if no ice is present, $h_s$ is set to zero. By combining Eqs. B11 and B24 the total melting of each ice layer is calculated as

$$\frac{dh_x}{dt} = \frac{dh_{x\_lower}}{dt} + \frac{dh_{x\_upper}}{dt}$$

(B25)

When $h_x < 0$ due to melting the surplus energy is used for melting neighbouring layers according to the following procedure: if the melting is initiated from above the surplus energy is used to melt the layer directly underneath; if the melting is caused by the water below the layer directly above receives the surplus melting energy; if $h_{ib} <= h_{iw} <= 0$ the water in the topmost grid cell is heated with the remaining energy.

**Ice model performance**

To test the ice module, Simstrat was calibrated in Sihlsee with PEST using monthly resolved vertical temperature profiles (2006 to 2008, RMSE 1.2 °C) for four parameters including the new p_albedo parameter for scaling snow/ice albedo. Modelled and monthly measured total ice cover from 2012 to 2018 is shown in Fig. B1 (RMSE 0.078 m). The modelled thickness agrees well with measurements during years with an extensive ice covered period (2013, 2014 and 2017, max height > 5 cm). The

5    model performance is not ideal for years with short temporal ice duration and thin ice thickness (2016 and 2018, max height < 5 cm). During these years, the quality of the forcing dataset becomes crucial. In the case of Sihlsee, the timing and duration of snowfall prolongs the duration of the ice-covered period. We use the meteorological station at Samedan (SAM) located four kilometres from the lake in a region with rapid topographical change. This in combination with monthly ice thickness measurements result in the divergence during 2016 and 2018.

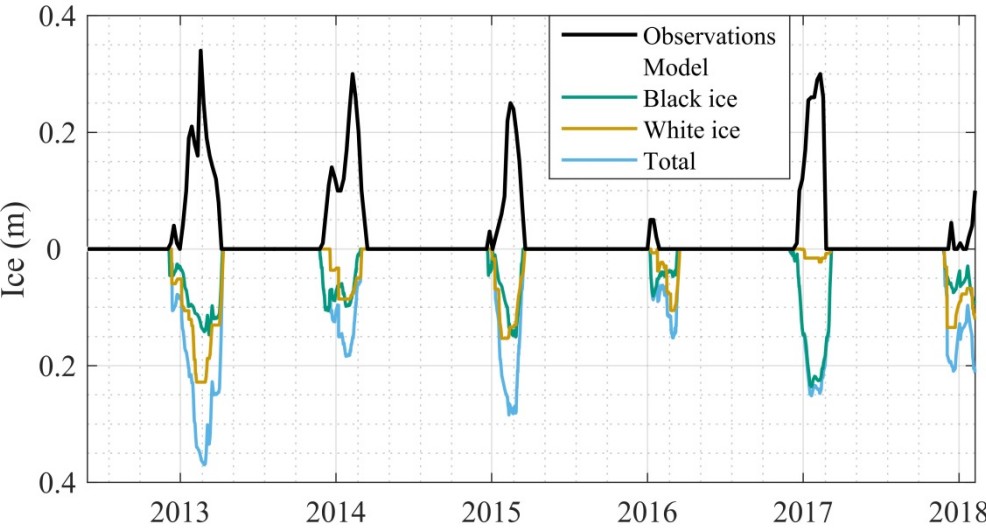

Figure B1. Ice model performance in Sihlsee (2012 to 2018) showing modelled white ice (orange), black ice (green) and total ice cover (white- and black ice combined, in blue) against measurements (black).

### C. Estimation of clear-sky solar radiation

The algorithm below is based on the equations from the Lake HeatFluxAnalyzer (see http://heatfluxanalyzer.gleon.org/), following the methods of Meyers and Dale (1983).

Declination of the sun [rad]: $\delta = \sin^{-1}(-0.39779 \cos \frac{2\pi DOY_s}{365.24})$, where $DOY_s$ is the day of year after the winter solstice (December 21$^{st}$).

Cosine of the solar zenith angle [-]: $\cos Z = \max(\sin \varphi \sin \delta + \cos \varphi \cos \delta \cos \frac{\pi(H-12.5)}{12}, 0)$, where $\varphi$ is the latitude in radians and H is the hour of the day, assuming the solar noon is at 12h30.

Air mass thickness coefficient [-]: $m = 35 \cos Z (1244 \cos^2 Z + 1)^{-0.5}$

Dew point temperature [°C]: $T_d = 243.5 \log \frac{p_w}{6.112} / (17.67 - \log \frac{p_w}{6.112}) + 33.8$, where $p_w$ [mbar] is the water vapour pressure.

Precipitable water vapour [cm]: $w_p = e^{0.1133 - \log(G+1) + 0.0393(1.8 T_d + 32)}$, where $G$ is an empirical constant dependent on latitude and day of year (see tables from Smith, 1966).

Attenuation coefficient for water vapour [-]: $\lambda_w = 1 - 0.077 (w_p m)^{0.3}$

Attenuation coefficient for aerosols [-]: $\lambda_a = 0.935^m$

Attenuation coefficient for Rayleigh scattering and permanent gases [-]: $\lambda_{Rg} = 1.021 - 0.084(m (0.000949 p_a + 0.051))^{0.5}$, where $p_a$ [mbar] is the air pressure.

Effective solar constant [W m$^{-2}$]: $I_{eff} = 1353(1 + 0.034 \cos \frac{2\pi DOY}{365.24})$, where DOY is the day of year.

Clear-sky solar radiation [W m$^{-2}$]: $H_{cs} = I_{eff} \cdot \cos Z \cdot \lambda_w \lambda_a \lambda_{Rg}$