# Peer review of "Toward an open-access of high-frequency lake modelling and statistics data for scientists and practitioners. The case of Swiss Lakes using Simstrat v2.1"

_Geoscientific Model Development, 2018_

## Referee Comment (RC1) · Anonymous Referee #1 · 2 May 2019

**Review of "Toward an open-access of high-frequency lake modelling and statistics data for scientists and practitioners. The case of Swiss Lakes using Simstrat v2.1", by Gaudard et al.**

In this study, the authors present a new open-access web-based platform with visualization and easy-access to simulations with the lake model Simstrat v2.1 for 54 lakes in Switzerland. The practical use of the platform is illustrated with two case studies, one to assess the effects of past climate change on the thermal structure of a lake, and second how short extreme events temporally affect the lake thermal structure. The presented platform is state-of-the-art but this might be stressed in the paper even more. Furthermore, the manuscript could benefit from some structural and textual changes, of which I included a list with suggestions under 'textual comments'. In general, the study can only be considered for publication if the comments specified here below are sufficiently addressed.

**General Comments**

1.  The main topic of the paper is to present the new online platform: I think this could be promoted even more throughout the paper:
    a.  The last paragraph of the introduction could be more elaborated. Also rewrite the sentence with 'with the intention of making our results openly accessible'. From what I understand, they are already open. More details could be provided on what is present on the platform. (In the introduction and/or in the results section, (P5 L13-15).
    b.  In the conclusion the main results of the two case studies as main advantages of the platform should be highlighted. I would also end the conclusion with a general statement about the platform.

2.  The manuscript could benefit from a slightly adjusted structure. Now, the results sections 3.1 and 3.2 describing the two case studies also include methodology and even literature review parts. Therefore I suggest to use a new structure as follows:

    2. Methods
    2.5. Case Studies
    2.5.1. Long-term evolution of the thermal structure of lakes: Lake Biel
    *Insert here paragraph 1 of page 6*
    2.5.2. Event based evolution of the lake thermal structure
    *Add here first paragraph of page 7*

**Specific comments**

1.  In the abstract, it would be good to specify that the lakes on the platform are modeled with one lake model, Simstrat. Also the sentences could be rephrased more directly. Some examples are included in the textual comments.

2.     P3 L19: 'an online platform': be more specific on which online platform: the new platform you present in this study? (see also general comment)

3.     Figure 1: Please make the titles of the input and output boxes consistent. I suggest to only use 'input' and 'output' (so remove the 'data' in 'input data'). Please apply the same consistency in the figure legend and caption.

4.     Figure 2: Please add color bar of lake temperatures and scale bar to figure. What is the green color on the figure representing? Please also add this in figure or figure caption.

5.     The authors state that 'inflows are disabled if no discharge or temperature data is available' (P4, L1). Is this the case for many lakes? Please identify the relevant lakes in Appendix table A and add the number in the text. Please also include a statement on the sensitivity of this methodological choice.

6.     P4 L2-5 and Appendix table A: please also indicate in the table for which lakes the Secchi depth measurements are available. Please also add a column with the lake tropic status, or provide the methodology of the classification in this paragraph.

7.     For the story continuation it is better to switch the third and second paragraph of P4. Like this, it makes more sense to first describe the timeframes and then how data gaps are treated. Please also take care of the transition in the data-gap paragraph.

8.     P4 L13-14: It is not clear to where the observations from the CTD profiles comes from. Please add the data source.

9.     P4 L17: please add more details on how the parameters for calibration were selected, at least include a reference of the previous sensitivity analysis.

10.    P4 L21: 'unless significant changes are made to either the model, forcing data or observational data'. In when is this the case? Please add more textual details on this.

11.    P4 L26: Please add the source of lake volume, temperature and densities.

12.    P5 L25-27: I would elaborate this paragraph, and discuss also the correlation coefficient showed in figure 3. Please also list the six lakes not shown in the figure caption.

13.    P5 L27: Please add more info to the study of Bruce et al., 2018: is it a global lake modelling study? Do they incorporate lakes in Switzerland as well?

14.     P6 L26-31: On line 26 there is indicated that a 'similar analysis' is done for all modelled lakes, however, only an inter-comparison of winter and summer stratification is showed and discussed, while in the case study for Lake Brienz, the trends in stratifications are investigated. Please rewrite the text to be consistent with the figures showed. Please add also more information on the possible implications of the delay of melt water runoff. Also, in the caption figure 6, there is no information on winter stratification, but on ice cover. Please update the text so that it is consistent with the information on the figure.

15.     Figure 7: Please remove X and Y labels, and add 'in Schmidt stability' to 'Delay/ Recovery time' colorbar caption.

**Textual comments**

These comments can just be implemented in the manuscript and do not have to be addressed in the response letter.

1.      P1 L11: Please replace 'hypothesizes' with 'hypotheses'
2.      P1 L13: please change 'openly accessible' to 'open-access' or synonym.
3.      P1 L15: Please change 'regional areas' to 'regions' and remove 'worldwide'
4.      P2 L2: please remove 'most'.
5.      P2 L5: Please change 'consists in' to 'consists of'
6.      P2 L6: Please change 'scarcely' to 'barely'
7.      P2 L7: Please change 'country-scale' to 'country-level'
8.      P2 L19: Please replace '(e.g. temperature profiles)' to ', like temperature profiles,'
9.      P2 L24: Please replace 'toolbox'
10.     P2 L24: Please replace 'properties' by 'variables'
11.     P2 L29: Please change 'more generally the public' to 'the public in general'.
12.     P3 L9: Please change 'clearer' to 'more clear'
13.     P3 L9: Please replace 'openly available' with 'freely available or other synonym
14.     P3 L10 Please replace 'refactoring' by a synonym
15.     P3 L15: Please add 'in the model' in 'the ice and snow module employed *in the model*, to enhance the sentence structure.
16.     P3 L18: Please add 'the' in '(iii) run *the* calibration' and on L19 'for *the* chosen model parameters'
17.     P3 L25: Please add 'the' in 'from *the* Federal Office of …'
18.     P3 L27: Please replace 'For hydrological forcing' by 'As hydrological forcing' or equivalent.
19.     P4 L7: Please replace 'depending on the data' by 'depending on the variable'
20.     P4 L6-11: Please revisit the whole paragraph and replace 'missing data' by 'data gaps' where appropriate. 'Missing data' can be interpreted as long series of data gaps.
21.     P4 L9: Please replace 'missing data are replaced' by 'data gaps are filled'
22.     P4 L11: 'The latter is calculated as described in Appendix C'. Here it is not clear that it points to the theoretical solar radiation. Please rephrase.
23.     P4 L18: Please replace 'maintain' with a synonym (e.g. 'keep').

24. P4 L29: Please change '.From this we calculate' to ', to calculate'
25. P5 L22: 'interfaced to' replace by synonym
26. P5 L23: Please rephrase sentence starting with 'Similarly, …'
27. P6 L4: Please replace 'were' with 'is'
28. P6 L18: Please change 'Contrarily' to 'In contrast to'
29. P6 L20: Please check figure numbers: two times 'Figure 4e and 4e'.
30. P6 L29: Add ', first reported by Livingstone et al. (2005), *which is* caused …'
31. Caption figure 2: 'Snaphop' to 'Snapshot'
32. Caption figure 6: Please add time period of data used in the figure and change last sentence to 'Lakes are ordered based on elevation form left (low elevation) to right (high elevation)' for clarity.
33. P7 L2: Please add 'a' in 'over *a* long period'.
34. At certain locations in the manuscript the language use is not entirely scientific and neutral. Here a list of this locations is provided, please change these to scientific wording or remove them:
    a. P1 L12 'by the modellers themselves'
    b. P2 L11: 'Although never perfect'
    c. P2 L15: 'Unfortunately'
    d. P2 L20: 'which remains a time-consuming process' and 'To be successful, such an endeavour'
    e. P2 L31: 'our extensive results'
    f. P5 L17: 'very well-suited'
    g. P7 L6: 'brutal', can be replaced by 'severe'
    h. P7 L12: 'by no means'
    i. P7 L22: 'obviously'
    j. P7 L29: 'comparatively overlooked'

16. P6 L7-31: In the whole section, the choice of words suggests that observations are analyzed, while model simulations are analyzed. Below some suggestions to improve this:
    a. L7: to the sentence 'we observe an increase in both yearly averaged surface and bottom temperatures' change to 'we observe an increase in both yearly averaged surface and bottom temperature *simulations*'
    b. L12: 'The vertical heterogeneous heating observed …' to 'the vertical heterogeneous heating modelled …'
    c. L14: 'We detect' to 'We simulate'
    d. Both L20 and L26: 'observed' to 'simulated'

17. P8 L15-18: Section *Code and Data Availability*: Please add here the used data sources as well, now the URLs are spread out over the document.
    a. Meteorological data from MeteoSwiss
    b. Hydrological forcing from the Federal Office for the Environment
    c. Data on CTD profiles (data source?)
    d. Geothermal data + URL (now in caption Table 2)
    e. Reference to PEST + URL (now on P4 L19)

---

## Referee Comment (RC2) · Anonymous Referee #2 · 9 May 2019

General comments:

Gaudard et al. presents a web-based platform for visualization and promotion of lake model outputs that are openly accessible to the general public. The web-based platform currently includes 54 lakes in Switzerland, and it could be useful in synthesizing lake model outputs in other geographical regions.

Specific comments:

Pg1, L13-14: and appropriate model, unless the authors have validated Simstrat v2.1

all over the world.

Pg2, L24: please replace 'It' with a real subject (e.g., model output data) to avoid potential interpretation confusion for this sentence.

Pg2, L26-27: 'and it can support the interpretation of biogeochemical observations, if the relevant processes are driven by thermal stratification and mixing'. This is confusing- does it mean models cannot support the interpretation of biogeochemical observations if the relevant processes are NOT driven by thermal stratification and mixing?

Pg3, L26-27: please replace 'adiabatic vertical rate' with the commonly used 'adiabatic lapse rate'. What are the ranges of altitude difference between the lakes and the meteorological stations? Adiabatic lapse rate is not necessarily -6.5 C/km, so such assumption could result significant errors when the altitude difference is large.

Pg4, L3-5: any reference that supports the light absorption coefficient parameterization described here?

Pg4, L8-10: what's the gap size for the 'highly seasonal variables'? How large is the inter annual variability for the 'highly seasonal variables', based on available measurements?

Pg5, L7: how do the authors determine the existence of ice? Is it measured or modeled?

Pg5, L25: what is the model validation period for RMSE? Is it the model timeframe listed in appendix A?

Pg 5, L 26: how large were the overestimations in the 6 lakes with RMSE > 2C?

Pg 6, L9: is the 'surface temperature' air temperature at the surface or lake surface temperature? Could the authors plot measured air temperature in Figure 4a?

Pg 6, L20: Figures 4e and 4e

P12, Fig1: please provide the full names of each abbreviation, e.g., what are Swisstopo, CTD, FOEN? Some abbreviations are defined in the main text (but scattering around), and it would be very helpful to list them in the figure caption. Also, observation files should be listed as an intermedium product instead of an output.

---

## Referee Comment (RC3) · Anonymous Referee #3 · 9 May 2019

Review of "**Toward an open-access of high-frequency lake modelling and statistics data for scientists and practitioners. The case of Swiss Lakes using Simstrat v2.1**", by Gaudard et al.

**General Comments**

In this paper the authors describe the development of an openly accessible web-based platform for visualization and data access of 54 lakes modelling in Switzerland. The lake modelling is conducted with a one-dimensional lake model Simstrat v2.1, which is the core scientific component of this paper. The other important component of this paper is the lake modelling platform, which is beneficial to both the general public and researchers. It is good that both components are included in this study; nevertheless, both components are not thoroughly introduced. As a scientific publication, higher portion of new scientific modules in Simstrat v2.1 and using Simstrat v2.1 for the scientific findings in a single event or from long-term climatic trends can benefit this paper.

**Specific Comments**

1.  The drawback of one-dimensional lake model is the lack of water circulation; nevertheless, the thermal dynamic in the lake can be very different from small lake to large one. Surface of the 54 studied lakes ranges from 0.102-km$^2$ of Lake Inkwilersee to 580-km$^2$ of Lake Geneva, which are quite diverse in horizontal dimension. It is not mentioned in the paper about the limitations and differences of applying one-dimensional Simstrat v2.1 to small and large lakes.
2.  In this study, four parameters among 46 lakes were calibrated. Now only the temperatures of post-calibration root mean square error were described. It would be good to summarize the calibrating processes, and the physical meanings of the calibrated parameters and its relationship to lake area and lake characters.
3.  P4, L1~5: In this study, the light absorption coefficient plays an important role determining incoming heat flux. Is there any reference, except current cited one (Poole and Atkins, 1929), using similar parameterization?
4.  P4, L6: What is the percentage of the missing forcing data in this study? And what is the impact of discrepancy in the model?
5.  P4, L10~11: It is not clear how the variable "cloud coverage" is used in the model, as the measured solar radiation is available.

6. P4, L13~14: Are all the lakes initialized for temperature and salinity using CTD profiles?

7. P5, L13: Why the platform is automatically updated with a weekly frequency?

**Textual Comments**

1. P4, L27: Missing a comma "," between the heat capacity of water and the volume of the lake.

---

## Author Comment (AC1) · 28 Jun 2019

In this study, the authors present a new open-access web-based platform with visualization and easy-access to simulations with the lake model Simstrat v2.1 for 54 lakes in Switzerland. The practical use of the platform is illustrated with two case studies, one to assess the effects of past climate change on the thermal structure of a lake, and second how short extreme events temporally affect the lake thermal structure. The presented platform is state-of-the-art but this might be stressed in the paper even more. Furthermore, the manuscript could benefit from some structural and textual changes, of which I included a list with suggestions under 'textual comments'. In general, the study can only be considered for publication if the comments specified here below are sufficiently addressed

We thank Reviewer 1 for his comments. We agree to stress more that the web-based one-dimensional hydrodynamic platform is state-of-the-art. This is now better stressed in the abstract, the introduction and the conclusion. We have also applied the structural and editing changes requested and thank the reviewer for this and took the opportunity of this review to extensively rework the manuscript.

General Comments

1. The main topic of the paper is to present the new online platform: I think this could be promoted even more throughout the paper:

a. The last paragraph of the introduction could be more elaborated. Also rewrite the sentence with 'with the intention of making our results openly accessible'. From what I understand, they are already open. More details could be provided on what is present on the platform. (In the introduction and/or in the results section, (P5 L13-15).

We have rewritten the last paragraph of the introduction. It now reads: "In this work, we present a new automated web-based platform to visualize and distribute the near real time (weakly) output of the one-dimensional hydrodynamic lake model Simstrat through an user-friendly web interface. The current version includes 54 Swiss lakes covering a wide range of characteristics from very small volume such as Inkwilersee (9 x $10^{-3}$ km$^3$) to very large systems such as Lake Geneva (89 km$^3$), over an altitudinal gradient (Lago Maggiore at from 193 m. a.s.l. to Daubensee at 2207 m. a.s.l.) and over all trophic states (14 euthrophic lakes, 10 mesotrophic lakes and 21 oligotrophic lakes, Appendix A). We focus here on describing the fully automated workflow, which simulates the thermal structure of the lakes and weekly updates the online platform (https://simstrat.eawag.ch) with metadata, plots and downloadable results. This state-of-the-art framework is not restricted to the currently selected lakes and can be applied to other systems or at global scale."

We have restructured the section 2.4 and the last paragraph was extended and moved to the beginning of the section. We also now provide more details on what is present on the platform

 b. In the conclusion the main results of the two case studies as main advantages of the platform should be highlighted. I would also end the conclusion with a general statement about the platform.

We have modified the conclusion to better reflect the results from the case studies: "We demonstrated the benefit of the platform through two simple case studies. First, we showed that the high frequency modelled temperature data allows a complete assessment of the effect of climate change on the thermal structure of a lake. We specifically show the need to evaluate changes in all atmospheric forcing, in the watershed or through-flow heat energy and in light penetration to accurately assess the evolution of the lake thermal structure. Then we showed that the high frequency modelled data can be used to investigate special events such as wind storms, there in-situ measurements under current temporal resolution are failing. ".

We have also added a more general statement regarding the platform at the end of the conclusion with the following sentences "By promoting a cross-exchange of expertise through openly sharing of in-situ and model data at high frequency, this open-access data platform is a new path forward for scientists and practitioners. "

2. The manuscript could benefit from a slightly adjusted structure. Now, the results sections 3.1 and 3.2 describing the two case studies also include methodology and even literature review parts. Therefore I suggest to use a new structure as follows:

2. Methods

2.5. Case Studies

2.5.1. Long-term evolution of the thermal structure of lakes: Lake Biel Insert here paragraph 1 of page 6

2.5.2. Event based evolution of the lake thermal structure Add here first paragraph of page 7

We agreed that the case studies should be introduced in the method section. We have added a subsection 2.5 Case studies where we briefly present the 2 case studies.

Specific comments

1. In the abstract, it would be good to specify that the lakes on the platform are modeled with one lake model, Simstrat. Also the sentences could be rephrased more directly. Some examples are included in the textual comments.
   The model is indicated in the title and as website. We do not think it is necessary to repeat the information.

2. P3 L19: 'an online platform': be more specific on which online platform: the new platform you present in this study? (see also general comment)
   Changed to" update the simstrat.eawag.ch online data platform to display"

3. Figure 1: Please make the titles of the input and output boxes consistent. I suggest to only use 'input' and 'output' (so remove the 'data' in 'input data'). Please apply the same consistency in the figure legend and caption.
   We have modified the figure and the caption accordingly

[Figure]

[Figure]

**Figure 1. General workflow diagram.** Model input (left box) is retrieved and processed by the Python script "Simstrat.py", which runs the model (Simstrat v2.1) and/or model calibration (using PEST v15.0) and produces output (right box). This output is then uploaded to a web interface (https://simstrat.eawag.ch) for general use. All scripts and programs are available on https://github.com/Eawag-AppliedSystemAnalysis/Simstrat/releases/tag/v2.1 and https://github.com/Eawag-AppliedSystemAnalysis/Simstrat-WorkflowModellingSwissLakes. Simstrat = one dimensional hydrodynamic model; CTD = Conductivity, Temperature, Depth profiler; PEST = Model independent parameter estimation and uncertainty analysis software; FOEN = Swiss Federal Office of Environment; MeteoSwiss = Swiss Federal Office of Meteorology and Climatology; Swisstopo = Swiss Federal Office of Topography

4. Figure 2: Please add color bar of lake temperatures and scale bar to figure. What is the green color on the figure representing? Please also add this in figure or figure caption.
   We now use the same color for each lake. The legend of the map is indicated as a link in the caption. We also now have indicated the locations of the 54 lakes.

[Figure]

**Figure 2. Illustration of the interactive map displayed on the homepage of the online platform: https://simstrat.eawag.ch. The location of the lakes discussed in this manuscript is also indicated with numbers (See Appendix A). Basemap is provided by Swisstopo and the specific legend can be found here https://api3.geo.admin.ch/static/images/legends/ch.swisstopo.swisstlm3d-karte-farbe_en_big.pdf**

5. The authors state that 'inflows are disabled if no discharge or temperature data is available' (P4, L1). Is this the case for many lakes? Please identify the relevant lakes in Appendix table A and add the number in the text. Please also include a statement on the sensitivity of this methodological choice.

   We have modified the Appendix A to better indicate this. We also added the theoretical residence time when data are available. In all low altitude lakes, lakes where the discharge is not accounted for are lakes with very weak inflows/outflows and large retention time. The influence on the thermal structure is therefore minimal. The problem is potentially larger for small high altitude lakes and should be further investigated in the future. Missing inflows and more generally watershed data is a source of error in small alpine lakes, yet, such error can be compensated during the calibration process. We have modified the text accordingly: "The aggregated discharge is the sum of the discharge of all inflows, and the aggregated temperature is the weighted average of the inflows for which temperature is measured. Inflow data are often missing for small or high altitude lakes (Appendix A). Missing inflows and more

generally watershed data is a source of error in small alpine lakes, yet, such error can be compensated during the calibration process."

6. P4 L2-5 and Appendix table A: please also indicate in the table for which lakes the Secchi depth measurements are available. Please also add a column with the lake tropic status, or provide the methodology of the classification in this paragraph.
We have added a new column regarding the trophic state and explicitly indicated the lakes with observed secchi depth information

7. For the story continuation it is better to switch the third and second paragraph of P4. Like this, it makes more sense to first describe the timeframes and then how data gaps are treated. Please also take care of the transition in the data-gap paragraph.
We have reversed and then merged and finally slightly extended the paragraph:
"The timeframe of the model is determined by the availability of the meteorological data (air temperature, solar radiation, humidity, wind, precipitation). Initial conditions for temperature and salinity are set using conductivity-temperature-depth (CTD) profiles or using the temperature information from the closest lake. We apply different data patching methods to remove data gaps from the forcing depending on the length of the data gap. For small data gaps with duration not exceeding one day, the dataset is linearly interpolated. In total < 1 % of the dataset is corrected using this approach. Longer data gaps of up to 20 days are replaced by the long-term average values for the corresponding day of the year. Only ~ 1.5 % of the dataset is corrected using this approach"

8. P4 L13-14: It is not clear to where the observations from the CTD profiles comes from. Please add the data source.
All the data source are provided as a link to the online platform in the acknowledgment

9. P4 L17: please add more details on how the parameters for calibration were selected, at least include a reference of the previous sensitivity analysis.
We added the following text to the Calibration section
"Model parameters are set to standard default values, and four of them are calibrated (see Table 2). The parameters p_radin and and f_wind scale the incoming long-wave radiation and the wind speed, respectively, and can be used to compensate for systematic differences between the meteorological conditions on the lake and at the closest meteo station. The parameter a_seiche determines the fraction of wind energy that feeds the internal seiches. This parameter is lake-specific, as it depends on the lake's morphology and it's exposition to different wind directions. Finally, the parameter p_albedo scales the albedo of ice and snow applied to incoming shortwave radiation, which depends on the ice/snow cover properties and is unknown for the individual lakes. The calibration parameters were selected according to their importance for the model (e.g. based on previous sensitivity analysis), and their number was deliberately kept small in order to keep the calibration process simple and focused. Calibration is

performed using PEST v15.0 (see http://pesthomepage.org), a model-independent parameter estimation software (Doherty, 2016)."

10. P4 L21: 'unless significant changes are made to either the model, forcing data or observational data'. In when is this the case? Please add more textual details on this.
We added the following information to the text: "e.g. release of a new version of Simstrat or delivery of a large amount of new observational data"

11. P4 L26: Please add the source of lake volume, temperature and densities.
Lake volume are extracted from Swisstopo the Swiss Federal Office of Topography, In situ observations comes from cantonal agencies or organisation such CIPEL (for Lake Geneva). They are indicated in the acknowledgment and fully listed on the web-based platform

12. P5 L25-27: I would elaborate this paragraph, and discuss also the correlation coefficient showed in figure 3. Please also list the six lakes not shown in the figure caption.
The six lakes with too large RMSE are now indicated with the symbol "°" in Figure 3.

We also discussed slightly more the model performance shown on Figure 3:
"The correlation coefficient remains always higher than 0.93 suggesting also that the model successfully reproduce the thermal structure of the investigated lakes. Overall, the quality of the results is better for lowland lakes than for high altitude lakes where local meteorological and watershed information are often missing."

13. P5 L27: Please add more info to the study of Bruce et al., 2018: is it a global lake modelling study? Do they incorporate lakes in Switzerland as well?
We have modified the text and added the following information: "This is comparable to the RMSE range of ~0.7-2.1 °C reported in a recent global 32-lake modelling study using GLM (Bruce et al., 2018) also including Lake Geneva, Lake Constance and Lake Zurich."

14. P6 L26-31: On line 26 there is indicated that a 'similar analysis' is done for all modelled lakes, however, only an inter-comparison of winter and summer stratification is showed and discussed, while in the case study for Lake Brienz, the trends in stratifications are investigated. Please rewrite the text to be consistent with the figures showed. Please add also more information on the possible implications of the delay of melt water runoff. Also, in the caption figure 6, there is no information on winter stratification, but on ice cover. Please update the text so that it is consistent with the information on the figure.
We agree with the Reviewer that we actually do not show the same analysis for all modelled lakes. This analysis cannot be summarized in 1 page in this manuscript and we have reformulated this statement accordingly. We also have modified the text regarding ice coverage and not inverse stratification as previously written. Note also that Figure 6 is now Figure 5. We removed the previous Figure 5 that was not necessary for this manuscript. The modified text related to this change is:

"Such analyses can be extended to all modelled lakes. An inter-comparison of the temporal extent of summer stratification and winter ice cover period is illustrated in Figure 5."

15. Figure 7: Please remove X and Y labels, and add 'in Schmidt stability' to 'Delay/ Recovery time' colorbar caption.
Modified

---

## Author Comment (AC2) · 28 Jun 2019

General comments:

Gaudard et al. presents a web-based platform for visualization and promotion of lake model outputs that are openly accessible to the general public. The web-based platform currently includes 54 lakes in Switzerland, and it could be useful in synthesizing lake model outputs in other geographical regions.

We thank Reviewer 2 for his comments

Specific comments:

Pg1, L13-14: and appropriate model, unless the authors have validated Simstrat v2.1

Simstrat v2.1 is validated for mid latitudes lakes and previous version were used in tropical lakes. We believe that this model can be used at global scale

Pg2, L24: please replace 'It' with a real subject (e.g., model output data) to avoid potential interpretation confusion for this sentence.

The sentence has been modified:

"Yet, model output data should not only be seen as a tool for temporal interpolation of measurements. Models also provide data of hard to measure quantities which are helpful for specific analyses (e.g., the heat content change to assess impact of climate change, or the vertical diffusivity to estimate vertical turbulent transport). Models finally support the interpretation of biogeochemical processes which often depend on the thermal stratification, mixing and temperature"

Pg2, L26-27: 'and it can support the interpretation of biogeochemical observations, if the relevant processes are driven by thermal stratification and mixing'. This is confusing- does it mean models cannot support the interpretation of biogeochemical observations if the relevant processes are NOT driven by thermal stratification and mixing?

Our model is a physical model for temperature, stratification and mixing in lakes. It is therefore correct that it can only help interpreting biogeochemical processes, if they are influenced by the physical processes. However, most biogeochemical processes in lakes are to some extent influenced by stratification, mixing and/or temperature. To clarify this, we modified the sentence to:

"Models finally support the interpretation of biogeochemical processes which often depend on the thermal stratification, mixing and temperature".

Pg3, L26-27: please replace 'adiabatic vertical rate' with the commonly used 'adiabatic lapse rate'. What are the ranges of altitude difference between the lakes and the meteorological stations? Adiabatic lapse rate is not necessarily -6.5 C/km, so such assumption could result significant errors when the altitude difference is large.

We have modified the sentence and have added a table indicating the altitude and coordinate of all meteorological stations used in this study. The difference is typically

O(10m) for low land lakes but this difference is indeed large for high alpine lakes like Lake Ritom and Lake Cadagno (~1000 m of altitude difference compared to the meteorological station), Daubensee (~ 800 m of altitude difference). We now indicate in the manuscript that meteorological station near high altitude lakes would be needed. "This correction is a source of error in high altitude lakes like Daubensee for which dedicated meteorological station would be needed."

Pg4, L3-5: any reference that supports the light absorption coefficient parameterization described here?

We refrain to refer to all papers providing secchi disk information on a Swiss lake. We used one already cited reference (Schwefel et al. 2016)

Pg4, L8-10: what's the gap size for the 'highly seasonal variables'? How large is the inter annual variability for the 'highly seasonal variables', based on available measurements?

We have rewritten the paragraph as follow: "The timeframe of the model is determined by the availability of the meteorological data (air temperature, solar radiation, humidity, wind, precipitation). Initial conditions for temperature and salinity are set using conductivity-temperature-depth (CTD) profiles or using the temperature information from the closest lake. We apply different data patching methods to remove data gaps from the forcing depending on the length of the data gap. For small data gaps with duration not exceeding one day, the dataset is linearly interpolated. In total < 1 % of the dataset is corrected using this approach. Longer data gaps of up to 20 days are replaced by the long-term average values for the corresponding day of the year. Only ~ 1.5 % of the dataset is corrected using this approach."

Pg5, L7: how do the authors determine the existence of ice? Is it measured or modeled?

The existence of ice is modeled. The model presented in Appendix B has been calibrated for Swiss lakes based on in situ observation of ice cover.

Pg5, L25: what is the model validation period for RMSE? Is it the model timeframe listed in appendix A?

Indeed. We have decided to use the entire time series and do not split between a calibration and a validation period. This could be done for the lakes having long time series of observations but reduce the accuracy of the calibrated parameters for shorter time series of observations

Pg 5, L 26: how large were the overestimations in the 6 lakes with RMSE > 2C?

We have modified the sentence as follow: "Out of the 46 calibrated lakes, the post-calibration root mean square error (RMSE) is < 1 °C for 17 lakes, between 1 and 1.5 °C for 15 lakes, between 1.5 and 2 °C for 8 lakes and between  2 °C and 3°C for 6 lakes (Figure 3), calibration data was not sufficient for 8 lakes in which we used standard settings."

Pg 6, L9: is the 'surface temperature' air temperature at the surface or lake surface temperature? Could the authors plot measured air temperature in Figure 4a?

We thank the reviewer for this comment that helped to rethink the Figure 4. We now also indicated other temperatures such as air temperature, total lake temperature and tributaries temperature. We show that the lake surface temperature (not the entire lake) is warming at a faster rate than the air temperature and discuss this in section 3.1.

[Figure]

Figure 4. Evolution of several indicators for Lake Brienz over the period 1981-2018; all linear regression have p_values << 0.001: (a) yearly mean lake surface temperature (0.74 °C/decade), yearly mean air temperatures (0.50 °C/decade), yearly mean tributary temperatures (0.26 °C/decade), yearly mean lake temperatures (0.22 °C/decade) and yearly mean bottom temperatures (0.16 °C/decade), with linear regression, (b) contour plot of the linear temperature trend through depth and month, (c) yearly start (+3.7 days/decade) and end (-7.5 days/decade) day of summer stratification, with linear regression, (d) yearly mean (line), min and max (shaded area) Schmidt stability, with linear regression, (e) yearly maximum Brunt-Väisälä frequency ($3.3 \times 10^{-4}$ $1/s^2$/decade), with linear regression (f) yearly mean (line), min and max (shaded area) heat content.

Pg 6, L20: Figures 4e and 4e

Modified

Discussion paper P12, Fig1: please provide the full names of each abbreviation, e.g., what are Swisstopo, CTD, FOEN? Some abbreviations are defined in the main text (but scattering around), and it would be very helpful to list them in the figure caption. Also, observation files should be listed as an intermedium product instead of an output.

We have modified Figure 1 as well as the caption

---

## Author Comment (AC3) · 28 Jun 2019

General Comments

In this paper the authors describe the development of an openly accessible web-based platform for visualization and data access of 54 lakes modelling in Switzerland. The lake modelling is conducted with a one-dimensional lake model Simstrat v2.1, which is the core scientific component of this paper. The other important component of this paper is the lake modelling platform, which is beneficial to both the general public and researchers. It is good that both components are included in this study; nevertheless, both components are not thoroughly introduced. As a scientific publication, higher portion of new scientific modules in Simstrat v2.1 and using Simstrat v2.1 for the scientific findings in a single event or from long-term climatic trends can benefit this paper.

We thank Reviewer 3 for his/her comments. We have largely reworked the manuscript to better show how the web-based platform can be used for scientific purpose. This is mostly evident in the section 3.1

Specific Comments

1. The drawback of one-dimensional lake model is the lack of water circulation; nevertheless, the thermal dynamic in the lake can be very different from small lake to large one. Surface of the 54 studied lakes ranges from 0.102-km2 of Lake Inkwilersee to 580-km2 of Lake Geneva, which are quite diverse in horizontal dimension. It is not mentioned in the paper about the limitations and differences of applying one-dimensional Simstrat v2.1 to small and large lakes.

The main limitation of 1D vertical model is that spatial variability is not accounted for. This is the reason why multibasin lakes like Lake Lucerne have been split into 4 different lakes characterised by distinct basin. This is the same for Lake Zurich, Lake Constance and Lake Lugano. We have written the following in the document: "For lakes with clearly defined multi basin such as Lake Lucerne, Lake Zurich, Lake Constance and Lake Lugano, each basin is considered as a separated lake connected to the other basins by inflows/outflows "

2. In this study, four parameters among 46 lakes were calibrated. Now only the temperatures of post-calibration root mean square error were described. It would be good to summarize the calibrating processes, and the physical meanings of the calibrated parameters and its relationship to lake area and lake characters.

We have added the following text:

"Model parameters are set to standard default values, and four of them are calibrated (see Table 2). The parameters $p\_radin$ and and $f\_wind$ scale the incoming long-wave radiation and the wind speed, respectively, and can be used to compensate for systematic differences between the meteorological conditions on the lake and at the closest meteo station. The parameter $a\_seiche$ determines the fraction of wind energy that feeds the internal seiches. This parameter is lake-specific, as it depends on the lake's morphology and it's exposition to different wind directions. Finally, the parameter $p\_albedo$ scales the albedo of ice and snow

applied to incoming shortwave radiation, which depends on the ice/snow cover properties and is unknown for the individual lakes. The calibration parameters were selected according to their importance for the model (e.g. based on previous sensitivity analysis), and their number was deliberately kept small in order to keep the calibration process simple and focused. Calibration is performed using PEST v15.0 (see http://pesthomepage.org), a model-independent parameter estimation software (Doherty, 2016)"

3. P4, L1~5: In this study, the light absorption coefficient plays an important role determining incoming heat flux. Is there any reference, except current cited one (Poole and Atkins, 1929), using similar parameterization?

The parameterization of the light absorption using a beer lamber law parameterized by one coefficient is the standard for limnological study. We added a more recent references (already used in the manuscript) to highlight this

4. P4, L6: What is the percentage of the missing forcing data in this study? And what is the impact of discrepancy in the model?

We have modified the text as follow:

"The timeframe of the model is determined by the availability of the meteorological data (air temperature, solar radiation, humidity, wind, precipitation). Initial conditions for temperature and salinity are set using conductivity-temperature-depth (CTD) profiles or using the temperature information from the closest lake. We apply different data patching methods to remove data gaps from the forcing depending on the length of the data gap. For small data gaps with duration not exceeding one day, the dataset is linearly interpolated. In total < 1 % of the dataset is corrected using this approach. Longer data gaps of up to 20 days are replaced by the long-term average values for the corresponding day of the year. Only ~ 1.5 % of the dataset is corrected using this approach"

5. P4, L10~11: It is not clear how the variable "cloud coverage" is used in the model, as the measured solar radiation is available.

Cloud coverage is needed for estimating incoming long wave radiation while solar radiation are needed for short wave radiation.

6. P4, L13~14: Are all the lakes initialized for temperature and salinity using CTD profiles?

Most lakes are initialized with data from CTD profiles. When not available, use information from the closest lake. The small discrepancy with the real temperature profile is quickly reduced (after < 6 months). The text was modified to better indicate this

7. P5, L13: Why the platform is automatically updated with a weekly frequency?

We did not think it was necessary to update it more frequently but already got multiple request to reduce the update frequency to the day. There is no technical obstacle but we prefer to work in improving the pipeline first.

Textual Comments

1. P4, L27: Missing a comma "," between the heat capacity of water and the volume of the lake.

modified

---

## Author Response (AR2)

Dear Editor,

Thanks for your interest in our manuscript. We have addressed the last points raised by Reviewer#2:

1. P4, L22: 'Model parameters are set to standard default values' Please remove "standard" in the sentence and the use of "default values" is sufficient.

Removed. See P4L22

2. P4, L24: Please replace 'meteo station' with the commonly used 'meteorological station'.

Modified See P4L24

3. P4, L27~28: The authors declare the parameter p_albedo 'is unknown for the individual lakes'. Four of the parameters are also "unknown" and calibrated. "p_albedo" should not be the only parameter addressed as "unknown". This sentence can be revised or removed.

Sentence revised:

"Finally, the parameter p_albedo scales the albedo of ice and snow applied to incoming shortwave radiation, which depends on the ice/snow cover properties."

See P4L27

4. P6, L5~6: Please be more specific about 'calibration data was not sufficient for 8 lakes in which we used standard settings'. It is not clear about the two words "sufficient" and "standard".

Sentence revised:

"There were too few in-situ observations on 8 lakes to perform a proper calibration and all parameters where thereby set to default values."

See P6L5

5. P6, L20~21: 'For the period 1981- 2015, the trend in solar radiation is 5 W/m2/decade'. Is it descending or ascending trend?

"Ascending" trend added. See P6L20

6. P6, L26: 'compared to other light'. Is it other lake or other light?

Modified: "compared to other lakes"

See P6L26

7. P6, L31: 'the inflows Aare and Lütschine'. Is it 'inflows of Aare and Lütschine'?

Modified: "for the inflows of the Aare and of the Lütschine rivers"

P6L31

8. P7, L6~8: The last sentence in this paragraph said 'The temporal change in the discharge and its temperature resulting from climate change should therefore be taken into account in predicting the change in lake thermal structure.' Do the authors try to express 'predicting future change in lake thermal structure'?

Modified: "The temporal change in the discharge and its temperature resulting from climate change should therefore be taken into account in studies attempting to predict the change in lake thermal structure."

See P7L6

9. P7, L28, L31: Please replace 'rate of warming' with the commonly used 'warming rate'.

Modified. See P7L28L31

10. P7, L29: Is it 'depends mostly to' or 'depends mostly on'?

Modified. See P7L29

11. P9, L4: Please replace 'open sharing' with 'openly sharing'.

Modified See P9L4